# Determinants of gastric cancer immune escape identified from non-coding immune-landscape quantitative trait loci

Christos Miliotis[1,2], Yuling Ma[1,3], Xanthi-Lida Katopodi [1], Dimitra Karagkouni[1,3,4], Eleni Kanata [1], Kaia Mattioli[1,6], Nikolas Kalavros [1,3,5], Yered H. Pita-Juárez [1,3], Felipe Batalini [1,7], Varune R. Ramnarine [1], Shivani Nanda [1,3,4], Frank J. Slack [1,4] ✉ & Ioannis S. Vlachos [1,3,4,5] ✉

The landscape of non-coding mutations in cancer progression and immune evasion is largely unexplored. Here, we identify transcrptome-wide somatic and germline 3′ untranslated region (3′-UTR) variants from 375 gastric cancer patients from The Cancer Genome Atlas. By performing gene expression quantitative trait loci (eQTL) and immune landscape QTL (ilQTL) analysis, we discover 3′-UTR variants with *cis* effects on expression and immune landscape phenotypes, such as immune cell infiltration and T cell receptor diversity. Using a massively parallel reporter assay, we distinguish between causal and correlative effects of 3′-UTR eQTLs in immune-related genes. Our approach identifies numerous 3′-UTR eQTLs and ilQTLs, providing a unique resource for the identification of immunotherapeutic targets and biomarkers. A prioritized ilQTL variant signature predicts response to immunotherapy better than standard-of-care PD-L1 expression in independent patient cohorts, showcasing the untapped potential of non-coding mutations in cancer.

Immune evasion is a key hallmark of tumorigenesis and cancer progression[1]. The central role of immune evasion in the development and maintenance of the disease is reflected in the remarkable efficacy of cancer immunotherapies[2]. The US Food and Drug Administration (FDA) recently approved the use of immune checkpoint inhibitors (ICIs), such as anti-PD-1 or PD-L1 monoclonal antibodies (mAb), as third-line treatments against advanced cancers of different types, including gastric cancer[3,4]. Despite the celebrated successes of cancer immunotherapy, only a small subset of patients still benefits, while the differences in clinical response even in tumors having similar histopathological types warrant the development of better biomarkers and means for patient stratification[5].

During tumor progression, cell populations with immune evasion properties, sometimes acquired through mutations, are clonally expanded[6]. The importance of coding germline and somatic mutations as drivers of immune escape has been extensively documented and is a domain of intense research[7]. However, less is known about the contribution of non-coding variants to immune evasion and tumorigenesis[8].

Messenger RNA 3′ Untranslated Regions (3′-UTRs) are primary sites for post-transcriptional regulatory events[9]. These processes account for ~60% of the variation in protein expression, while ~20% of germline expression quantitative trait loci (eQTLs) are located in 3′-UTRs[9], which are more conserved than other noncoding loci, suggesting selective pressure[10]. 3′-UTRs are the most common targets of

[1]Harvard Medical School Initiative for RNA Medicine, Department of Pathology, Beth Israel Deaconess Medical Center, Harvard Medical School, Boston, MA, USA. [2]Harvard Program in Virology, Harvard University Graduate School of Arts and Sciences, Boston, MA, USA. [3]Broad Institute of MIT and Harvard, Cambridge, MA, USA. [4]Cancer Center & Cancer Research Institute, Beth Israel Deaconess Medical Center, Harvard Medical School, Boston, MA, USA. [5]Spatial Technologies Unit, Beth Israel Deaconess Medical Center, Boston, MA, USA. [6]Present address: Division of Genetics, Department of Medicine, Brigham and Women's Hospital and Harvard Medical School, Boston, MA, USA. [7]Present address: Division of Oncology, Department of Medicine, Mayo Clinic, Phoenix, AZ, USA. ✉e-mail: fslack@bidmc.harvard.edu; ivlachos@bidmc.harvard.edu

key regulatory molecules such as RNA binding proteins (RBPs) and microRNAs (miRNAs).

miRNAs are potent post-transcriptional regulators and are implicated in the control of numerous cellular mechanisms[11] as well as of all cancer hallmarks[12], hence their role in cancer immune surveillance has become a research hotspot[12]. miRNAs have been found to efficiently regulate Programmed death ligand 1 (PD-L1), other B7 family members, cytokines and numerous immune genes[13,14]. On the other hand, RBPs have been shown to regulate mRNA processing, localization, interactions, and stability[15], while different RBPs such as Mex3B[16], Mex3C[17], and HNRNPR[18] have been shown to regulate key antigen presentation mechanisms.

However, tumors mutate, truncate, or edit their 3′UTRs to escape this tight regulatory control[13,19–23]. Unfortunately, the current reliance of variant-calling pipelines on whole exome sequencing (WES) data, which do not include probes for 3′-UTR regions, has resulted in a lack of understanding of the role of 3′-UTR variants in cancer progression. Small-scale, targeted studies have identified individual 3′-UTR somatic mutations that associate with changes in *cis*-gene expression and immune phenotypes, especially for *PD-L1*[24–26]. For instance, a common somatic mutation in the *PD-L1* 3′-UTR has been shown to disrupt miR-570 binding leading to increased expression[26].

RNA sequencing (RNAseq) has been shown to be an alternative variant detection source[27]. The Pan-Cancer Analysis of Whole Genomes (PCAWG) database contains the largest collection of cancer patient samples with whole genome sequencing (WGS) data to date[28]. In PCAWG (*n* = 1188), out of the 87 samples without a driver alteration identified at the DNA level and available RNAseq data, every sample had an RNA-level alteration identified; indicating that driver alterations could have revealed themselves in RNA, rather than DNA[29]. Until today, no study has leveraged the rich transcriptomic data from The Cancer Genome Atlas (TCGA) to identify mutations across this vast resource.

Here, we perform a comprehensive mutational analysis on raw RNAseq data from hundreds of stomach adenocarcinoma (STAD) samples in TCGA to identify 3′-UTR germline and somatic single-nucleotide variants (SNVs) as well as short insertions-deletions (indels)[30]. By performing a quantitative trait loci (QTL) analysis, we identify *cis*-acting gene expression QTLs (*cis*-eQTLs), as well as variants associated with changes in immune phenotypes, herein termed immune landscape QTLs (ilQTLs). We design and implement a massively parallel reporter assay (MPRA) to validate at scale *cis*-eQTLs in immune-related genes directly affecting post-transcriptional stability and abundance of respective genes. MPRAs have been utilized successfully in the past for the functional validation of non-coding variants, such as promoter and UTR germline variants[31], while this assay is specifically enriched in somatic 3′-UTR mutations. We also investigate the translational potential of the identified 3′-UTR variants and specifically their ability to predict outcomes across diverse cohorts of ICI. Utilizing the prioritized ilQTLs, we establish a polygenic risk score (PRS) that proves more accurate in predicting response to checkpoint inhibition in melanoma and gastric cancer patients than PD-L1 expression, providing direct support of the potential utility of UTR variants in predictive modeling in immunotherapy. In this work, we establish the tools and apply them to unbiasedly identify transcriptome-wide 3′-UTR variants associated with changes in *cis*-gene expression and immune phenotypes in cancer and lay the foundations for similar 3′-UTR-focused studies in other cancer types.

## Results

### Characterization of the somatic and germline 3′-UTR variant landscape in gastric adenocarcinoma

Apart from the 1188 PCAWG samples overlapping with TCGA, WGS data are not available for ~90% of TCGA subjects, limiting large scale 3′-UTR variant investigations. Specifically for gastric cancer, only 40 samples comprise WGS data. The only study to date which

attempted to analyze 3′-UTR variants in TCGA in non-WGS samples[32], mistakenly considered that 3′-UTR regions were captured in the WES probe sets used in the study[30,33]. We now know that these regions are not covered in the TCGA WES kits[30], with only 0.31% of 3′UTR regions being targeted in the TCGA STAD cohort.

To call variants, we utilized RNAseq data, which has been shown to be a powerful modality for such analyses[27,28]. Sequencing data from 375 gastric cancer patients (Supplementary Data 1), including 375 primary gastric cancer samples and 40 matched controls, were analyzed following a comprehensive approach using GATK best practices[34] (Fig. 1A) and led to the identification of thousands of expressed variants and indels per sample (Supplementary Fig. 1A, B). Analysis of the distribution of called variants along the length of the 3′-UTR revealed that the RNAseq-derived calls matched the distribution of WGS calls from PCAWG (Fig. 1B). As expected, analysis of the distribution of TCGA WES-derived variants along the length of the 3′-UTR showed that most variants fall in the beginning of the 3′-UTR, proximal to coding sequences, likely representing sequences captured by coding region probes (Fig. 1B). The majority of 3′-UTR variants distally to the stop codon are missed by the WES-based variant calling analysis, due to lack of targeting probes.

Out of 5,431,118 variants identified across the genome by the RNAseq GATK analysis post-filtering, 3,283,340 (60.5%) overlapped with germline variant calls identified from blood samples from the same patients. The remaining 2,147,778 variants (39.5%) were treated as "likely somatic" calls. Of the likely somatic calls, 1,429,039 variants (66.5%) intersected with common RNA editing events identified in the GTEx database[35]. Samples deemed as ultramutated by TCGA, based on WES-derived variant calls, exhibited high frequency of RNAseq-derived somatic SNVs (Fig. 1C). We also reanalyzed all TCGA STAD RNAseq samples using Strelka2[36], an orthogonal variant calling algorithm, and identified a ~90% concordance (Fig. 1D).

### High-throughput capture of known functional variants in *PD-L1* 3′-UTR

We initially evaluated whether our high-throughput approach could capture the functional impact of the small number of 3′-UTR SNVs that have been previously associated with changes in *PD-L1* expression in gastric cancer, such as the polymorphisms *rs2297136*[25] and *rs4143815*[37], or non-small cell lung cancer, such as *rs4742098*[38]. Indeed, our analysis identified all three variants and when comparing *PD-L1* expression in samples with or without the 3′-UTR variants, we observed significantly increased expression levels in patients carrying the alternative allele, as expected based on the literature (Fig. 2).

### Prioritization of 3′-UTR germline variants and somatic mutations controlling *cis* gene expression in gastric adenocarcinoma

We prioritized transcriptome-wide 3′-UTR variants associated with *cis*-gene expression changes in gastric cancer. We investigated variants that were present in 5 or more samples, corresponding to 1.3% or higher minor allele frequency (MAF) in the tested population (2,917,776 total variants, 68,4%/1,994,516 germline, 31.6%/923,260 somatic, in which 67.2%/620,419 overlapping with editing sites). We performed an eQTL analysis[39] with the dosage of each variant as the genotype variable and the inverse quantile-normalized expression of the corresponding gene as the phenotype variable. To remove unwanted variation from the model, we used sex, age, along with the top 5 genetic principal components (PCs, Supplementary Fig. 2A) and expression surrogate variables (SVs)[40] as covariates (a simplified version of the formula is captured in Fig. 3A).

With a cutoff of a nominal *p* value of 1e-7, we identified ~3000 *cis*-eQTLs in protein-coding genes, accounting for 75% of all eQTLs (Fig. 3B). Out of the 3133 exonic (CDS/UTR) variants in protein coding genes, 1845/58.9% were germline, and 1288/41.1% were somatic

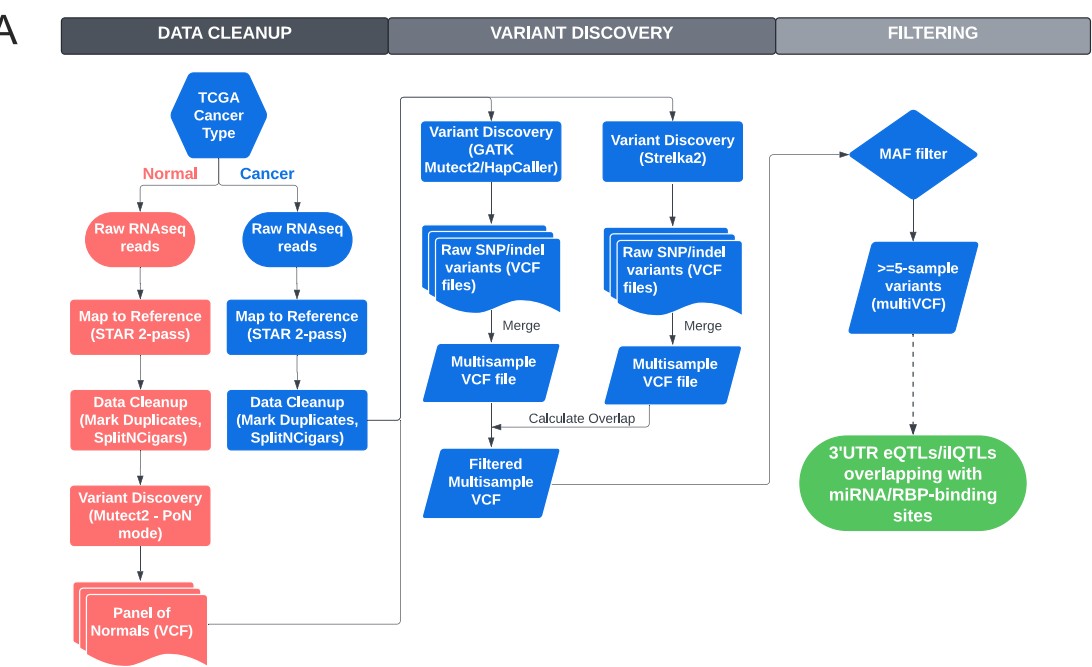

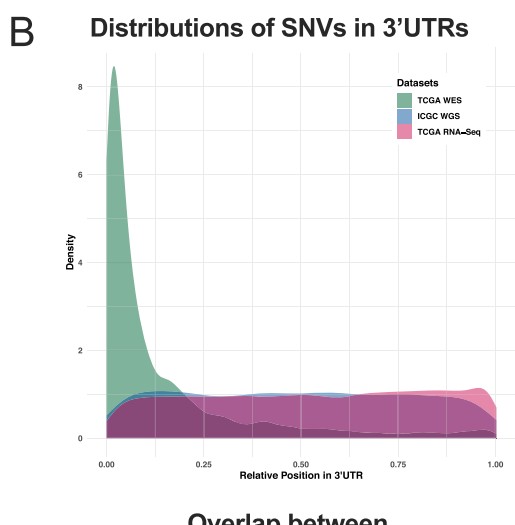

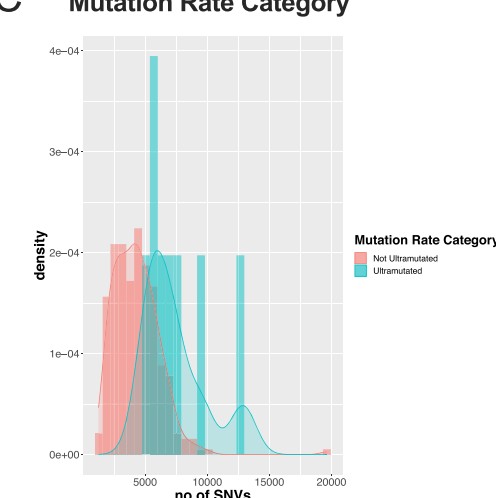

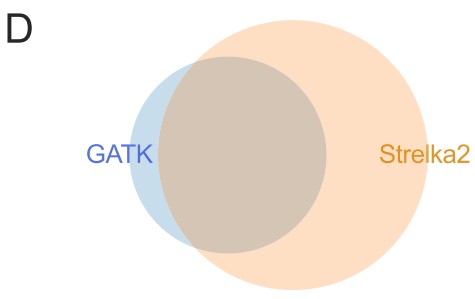

**Fig. 1 | Pipeline and quality control for identification of 3′-UTR variants in TCGA STAD cohort. A** Flowchart showing the variant-calling pipeline used for the analysis of TCGA STAD raw RNAseq data. **B** Distribution of the relative position of single nucleotide variant calls along the 3′-UTR as identified by analysis of TCGA WES, RNAseq and WGS data. **C** Number of RNAseq-derived somatic 3′-UTR single nucleotide variants per sample, grouped by "Mutation Rate Category" as defined by TCGA WES variant-calling data. **D** Overlap between transcriptome-wide variant calls in the analysis of RNAseq data with GATK and Strelka2 (GATK = 5,431,118, Strelka2 = 10,175,223, Overlap = 4,692,062). Figure data are provided in the Source Data file.

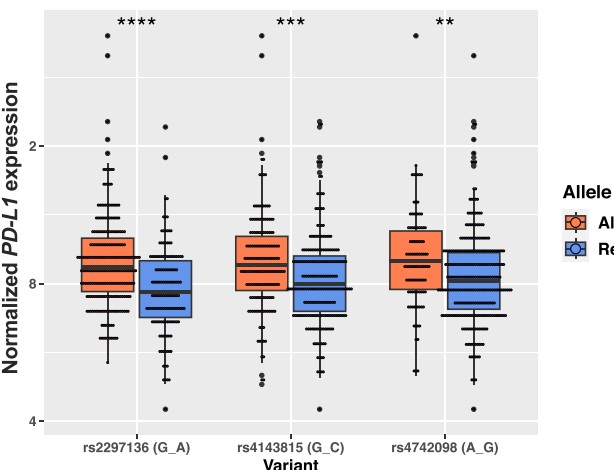

**Fig. 2 | Analysis of *PD-L1* 3′-UTR variants in TCGA STAD.** Normalized *PD-L1* expression (log2(read count+1)) in patients carrying the alternative allele (Alt) vs patients homozygous for the reference allele (Ref). For all three SNPs studied, Alt patients show higher levels of *PD-L1* expression than Ref patients. Boxplot lines represent the median and upper or lower quartiles, while whiskers define the 1.5x interquartile range. Significance was assessed by two-tailed Student's *t* test. rs2297136 (nAlt = 236, nRef = 139, *p* = 1.42e-8), rs4143815 (nAlt = 163, nRef = 212, *p* = 1.01e−4), rs4742098 (nAlt = 101, nRef = 274, *p* = 0.00173). Figure data are provided in the Source Data file. *P < 0.05, **P < 0.01, ***P < 0.001, ****P < 0.0001.

(254 overlapping with editing sites). 27.6% of the identified eQTLs overlapped with eQTLs from the Genotype-Tissue Expression project (GTEx), identified using healthy stomach tissue samples (90.6% overlapping germline eQTLs). Interestingly, we found that the highest percentage (60%) of all significant *cis*-eQTLs in protein-coding genes reside in the 3′-UTR genic region (Fig. 3C), reflecting the importance of 3′-UTR *cis*-acting elements in controlling gene expression, often post-transcriptionally[41]. Comparing the number of 3′-UTR variants to the average relative length of the 3′-UTR in protein coding genes[42] revealed a significant enrichment (Supplementary Fig. 2B, C) (chi-square *p* value < 1e-5). Enrichment of eQTLs in the 3′-UTR among exonic sequences has also been reported in other studies, including analyses of normal-tissue eQTL variants from GTEx[43,44]. Somatic eGenes, where the lead eQTL was somatic, included important gastric cancer oncogenes, such as *KRAS*, *CCDN1*, and *CCND2* among others[45], genes involved in antigen generation, processing, or presentation (e.g., *APOBEC3B*, *CANX*, *CTSS*), and cytokines/chemokines or other key immune regulators (e.g., *STAT1*, *CXCL5*, *CXCL9*, *TNFRSF9*). Using cell compartment marker sets derived from a gastric cancer single cell atlas[46], we observed that the eQTLs of somatic-only variants were enriched in tumor/epithelial cell markers (all somatic: Storey's *q* value = 1.19e-13, somatic-only: Storey's *q* value = 5.72e-8). This enrichment was diminished for germline-only eQTLs (Storey's *q* value = 0.025).

Overlapping the 3′-UTR *cis*-eQTLs with databases of predicted and experimentally supported miRNA and RBP binding sites showed that around 90% of the variants reside in functionally relevant regulatory elements (Fig. 3D)[11,47–49]. However, by performing a permutation-based test against randomly sampled 3′UTR regions, only predicted miRNA binding sites and not RBPs were identified as significantly enriched in the eQTL loci (*p* = 0.0002). Finally, gene-set enrichment analysis (GSEA) of the top significant 3′-UTR *cis*-eQTLs revealed enrichment in immune-related pathways (Fig. 3E), indicating that 3′-UTR variants could have an impact on immune phenotypes in gastric cancer.

## Massively parallel reporter assay validation of cis-eQTLs residing in cancer immunoediting genes

Considering the significant enrichment of immune-related pathways in the top 3′-UTR *cis*-eQTLs as well as the importance of immune escape in cancer progression and as a target for novel therapeutics, we proceeded with a functional validation of prioritized 3′-UTR eQTLs on a compiled list of immune-related genes (Supplementary Data 2, Methods). The manually curated list incorporates immune checkpoint and known or suspected immunomodulatory genes, MHC machinery, genes used in signatures for response to immunotherapy[50–53], and significant hits from hypothesis-free CRISPR–Cas9 screens for CD8+ T-Cell effector function[54,55] and in vivo screening of transplantable tumors in mice treated with ICI[56].

We selected the top 749 variants (478 somatic, 135 potential editing sites) that resided in 299 prioritized genes based on the curated list described above (Supplementary Data 3). We developed and performed a massive parallel reporter assay (MPRA) in two gastric cancer cell lines to assess the effect of each variant on the post-transcriptional stability of a reporter gene (Fig. 4A, Methods). Briefly, a reporter plasmid library containing barcoded reference and alternative alleles for the 749 eQTL variants was transfected into AGS and SNU719 cells. The effect of the variant on post-transcriptional expression of the reporter was assessed by barcode quantification from amplicon sequencing of RNA extracted from transfected cells.

The two cell lines used in the MPRA assay yielded similar outcomes (Supplementary Fig. 3A, B). Approximately 15% of eQTLs (128 variants) showed a significant causative effect on the expression of the reporter (FDR-adjusted *p* value < 0.05) in at least one of the two cell lines (Supplementary Data 4). A subset of genes with causal regulatory variants (Fig. 4B, Supplementary Data 4) are involved in antigen processing and presentation (e.g., HLA genes, *CTSB*, *CTSS*, *LGMN*, *CIITA*, *TAPBP*) as well as RNA-editing enzymes such as *ADAR*.

## Uncovering transcriptome-wide 3′-UTR variants regulating the gastric adenocarcinoma immune landscape

3′-UTR regions not only regulate gene expression, but can also influence mRNA localization, protein-protein interactions and other post-transcriptional, translational and post-translational functions[41]. Therefore, 3′-UTR variants can affect immune phenotypes in cancer independently of their effect on expression. To unbiasedly associate 3′-UTR variants with changes in the immune landscape of gastric cancer, we performed an immune landscape (il)QTL analysis using a similar model as above (Fig. 3A), focusing on immune phenotypes instead of *cis*-gene expression as the dependent variable. Immune phenotypes for the TCGA STAD cohort were obtained from the Cancer Research Institute (CRI) iAtlas project and included expression-based immune cell infiltration estimates, TCR/BCR entropy and leukocyte ratio calculated by combined imaging, methylation, and expression-based analyses[57].

A total of 1715 ilQTLs were identified, with 159 (9.3%) identified as germline. Among the remaining 90.7%/1,556 "likely somatic" ilQTLs, 370 intersected with common RNA editing events. For almost all immune features, the majority of ilQTLs resided in the 3′-UTR region of protein-coding genes and a large percentage of those were predicted to reside in regions of miRNA/RBP binding (Fig. 5), similarly to eQTLs. By performing the permutation analysis for CD8+ T cell ilQTLs, as in eQTLs, they were found to be enriched in predicted (*p* = 0.002), and experimentally validated (*p* = 0.005) miRNA binding sites as well as in experimentally supported RBP binding loci (*p* = 0.002). CD8+ T cell ilQTLs were also significantly more frequent to be of somatic origin, as compared to germline (Somatic CD8+ T cell ilQTLs: 95.58%, 2.3e-165, two-sided Fisher's exact test). Significant ilQTLs, the majority of which of somatic origin, were identified in immune-relevant genes, including *B2M*, HLA genes, *CANX*, *LDHA*, *PSMB2*, and *HNRNPR*, which are known to affect the tumor immune landscape[5,17,58–61]. The top hits also

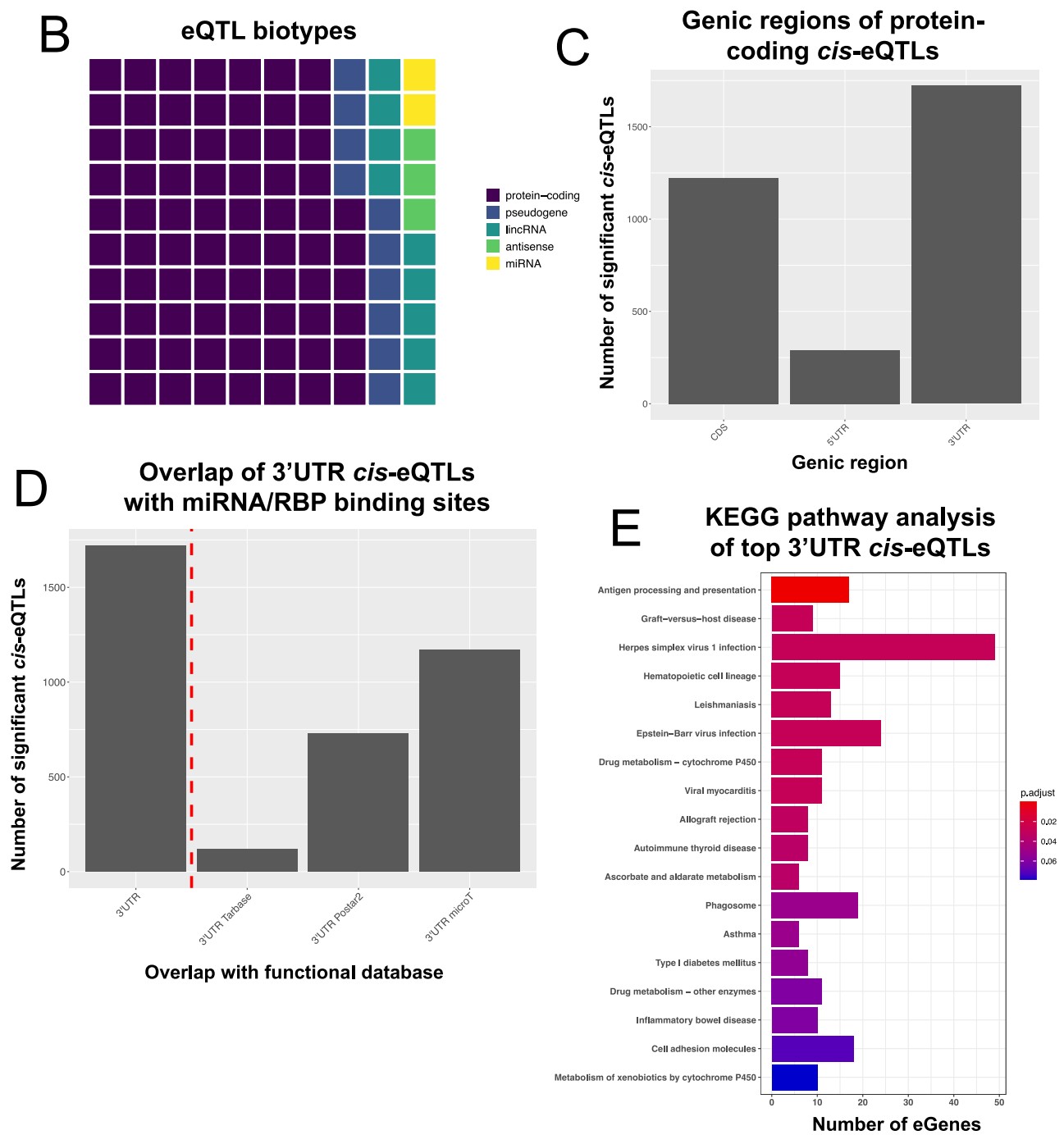

**Fig. 3 | Identification and characterization of 3'-UTR cis-eQTLs in TCGA STAD.**
**A** Model used for eQTL analysis. **B** Waffle plot showing the percentage of eQTLs that mapped to each of the indicated gene biotypes. A waffle plot consists of 100 squares and the number of colored squares represents the percentage of eQTLs mapping to each gene biotype. **C** Barplot showing the number of protein-coding cis-eQTLs that reside in each genic region (5'UTR, CDS or 3'-UTR). **D** Number of 3'-UTR cis-eQTLs that overlap with miRNA or RBP binding sites in the indicated databases. TarBase is a database of experimentally validated miRNA binding sites, microT includes predicted miRNA binding sites and POSTAR2 contains predicted RBP binding sites based on CLIP experimental data. **E** KEGG pathway enrichment analysis of the 500 topmost significant 3'-UTR cis-eGenes. The X axis shows the number of eGenes that belong to the indicated pathway and the color (legend) represents the FDR-adjusted one-sided Fisher's exact test p value for the enrichment of each gene-set. Figure data are provided in the Source Data file.

included *WARS*[62] and *APOBEC3C*[63], which were only recently implicated in tumor immunity, showing the potential of this approach for prioritization of novel cancer-specific immunotherapeutic targets.

We focused on CD8+ T cell ilQTLs since the level of CD8+ T cell infiltration in a tumor is an important determinant of cancer

immunotherapy response[64]. We identified significant CD8+ T cell fraction QTL variants in 467 genes. GSEA analysis showed that the most enriched "cellular component" gene sets in the topmost significant CD8+ T cell infiltration QTL variants are the ribosome and ribosomal subunits (Supplementary Fig. 4). In addition, immunoregulatory

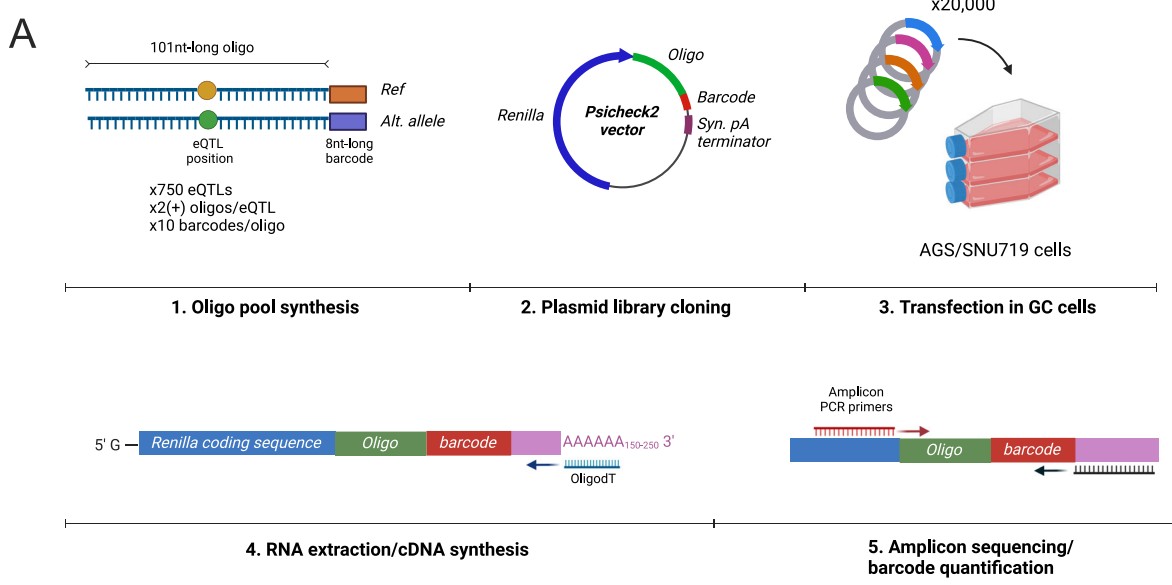

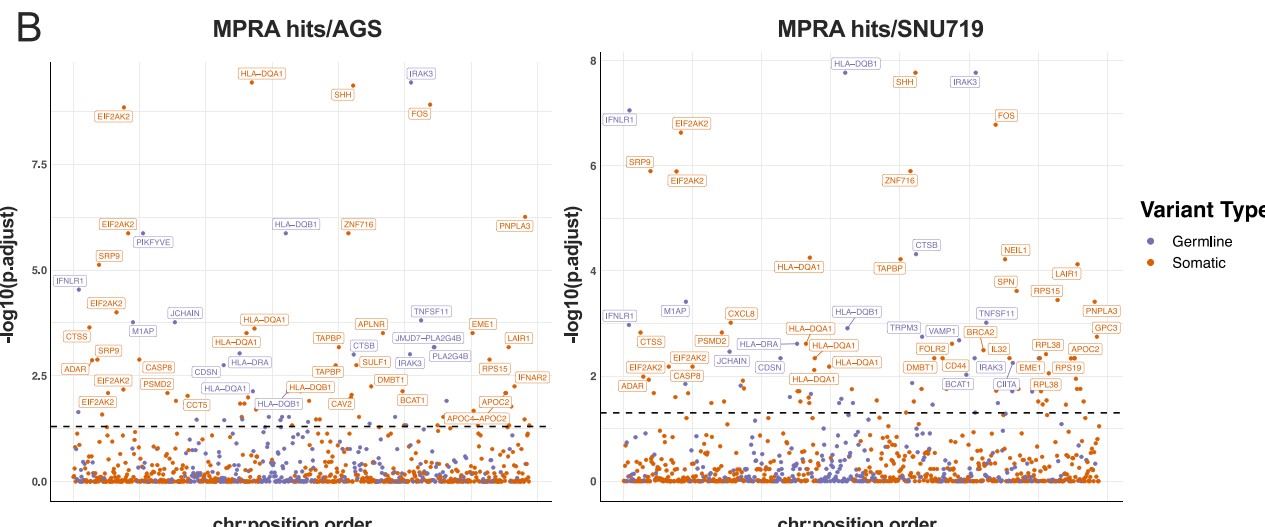

**Fig. 4 | Massively parallel reporter assay for the characterization of 3'-UTR immune-related cis-eQTLs. A** Protocol followed for 3'-UTR MPRA assay. Ref: Reference allele, Alt. allele: Alternative allele, Syn. pA terminator: synthetic polyadenylation terminator, oligodT: primer containing 20 T nucleotides. Created with Biorender.com. **B** Manhattan plot showing the top hits from the MPRA assay in both gastric cancer cell lines. A two-sided Wilcoxon rank sum test was used to calculate *p* value of enrichment of alternative vs reference alleles and multiple comparison adjustment was performed by FDR. The black line (*p*. adj. = 0.05) separates significant from non-significant calls. Blue dots represent likely germline variants, while red dots represent likely somatic variants. Figure data are provided in the Source Data file.

CRISPR hits (Supplementary Data 2) were enriched in significant CD8+ T cell ilQTL 3'-UTR variants compared to all 3'-UTR variants (1.42-fold increase, one-sided Fisher's exact test *p* value = 0.0057), showing concordance between the two orthogonal approaches of key gene prioritization. In addition, only 5% of those ilQTLs overlapped with eQTLs, showing the ability of ilQTLs to capture associations beyond gene expression regulation.

**Functional 3'-UTR cis-eQTL and ilQTL variants in *ADAR***
Significant functional variants were identified in all our high throughput investigations (eQTLs, ilQTLs, MPRA validated variants) in the *Adenosine Deaminase RNA Specific* (*ADAR*) gene. *ADAR* encodes an enzyme that catalyzes A-to-I editing in RNA and has been implicated in promoting cancer hallmarks in multiple cancer types, including breast, thyroid and gastric malignancies[65–67]. In the CRISPR screen by Manguso et al.[56] for genes that sensitize tumors to immunotherapy in mouse

models, *ADAR* is the 4th most enriched out of ~20,000 genes. *ADAR* downregulation has also been shown to induce inflammatory signaling in gastric cancer specifically[68]. To investigate the immunoediting role of *ADAR* in gastric cancer further, we queried the iAtlas portal[57], where CNVs on *ADAR* were reported to exhibit high effect sizes on Leukocyte Fraction (Amp: $p = 10^{-4}$ to $10^{-10}$ (multiple groups)), Lymphocyte Infiltration Score, (Amp: $p = 10^{-6}$/Del: $p = 10^{-5}$), and CD8+ T Cell content (Del: $p = 10^{-4}$). Moreover, *ADAR* harbored multiple 3'-UTR eQTL and ilQTL variants in gastric cancer (Fig. 6A), while we also found *ADAR* to be significantly overexpressed in responders (*n* = 55) compared to non-responders (*n* = 80) to immune-checkpoint inhibitors, in a cohort consisting of gastric cancer and melanoma patients (Fig. 6B)[50,51,69,70]. By meta-analyzing additional studies through the tumor immunotherapy gene expression resource (TIGER), we saw that *ADAR1* overexpression is a common feature for response to checkpoint inhibition (Supplementary Fig. 5). On the other hand, Manguso and colleagues showed

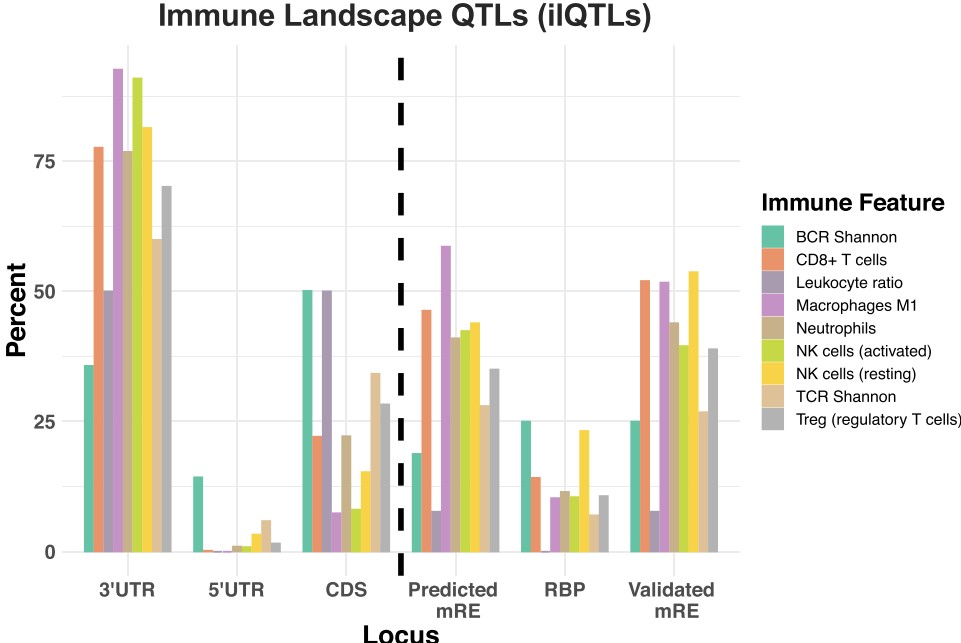

**Fig. 5 | Identification and characterization of 3′-UTR immune-related ilQTLs in TCGA STAD.** Left-hand side of dashed black line: distribution of significant ilQTLs along genic regions (5′UTR, CDS or 3′-UTR), showing enrichment in 3′-UTR variants. Right-hand side of dashed line: overlap of 3′-UTR ilQTLs with databases for miRNA response elements (mRE) and RBP binding sites. Figure data are provided in the Source Data file.

that loss of ADAR1 overcomes resistance to PD-1 checkpoint blockade caused by inactivation of antigen presentation by tumor cells in mouse models of resistance[59].

Through our QTL and MPRA analysis, we identified a somatic 3′-UTR *cis*-eQTL variant (chr1:154583325, T-to-C) in *ADAR* (Fig. 6A) with causal effects on post-transcriptional regulation (Supplementary Fig. 6A, B). *ADAR* variants including (chr1:154583325, T-to-C) were also found as ilQTLs for multiple immune features (Fig. 6A). Based on a meta-analysis of RNA cross-linking and immunoprecipitation (CLIP) data from the POSTAR2 project[49], this *ADAR* variant is predicted to overlap with multiple RBP (Supplementary Table 1) and miRNA (Supplementary Table 2) binding sites. One of those RBPs, TARDBP (Supplementary Table 3), has been studied for its ability to regulate gene expression, pre-mRNA editing, mRNA localization, and microRNA processing through binding on canonical GU-rich motifs or non-canonical sequences, with 3′UTRs being commonly targeted regions[71]. TARDBP has been shown to directly regulate *ADAR1* expression in liver cancer and leukemia cell line models[72]. Indeed, correlation analysis in gastric cancer patients from TCGA revealed a strong association in the expression of the two genes (Fig. 6C), suggesting that the regulation axis between TARDBP and ADAR could be functional in gastric cancer as well.

### 3′-UTR variants as an effective means for immunotherapy response prediction

We next sought to investigate the translational potential of 3′-UTR ilQTL variants and their ability to predict therapeutic outcomes to cancer immunotherapy. To this end, we analyzed the cohort of responders (R) vs non-responders (NR) to ICIs described above[50,51,69,70], where tumors were subjected to whole transcriptome sequencing thus enabling the detection of 3′-UTR variants.

We separated the samples into a training ($n = 68$, 39.7% Responders) and a test ($n = 67$, 40% Responders) set and selected 28 TCGA 3′-UTR ilQTL variants that were enriched in the R vs NR samples of the training set (one-sided Fisher's exact test $q$ value < 0.05, Supplementary Table 4). The variants were used to devise a Polygenic Risk Score (PRS) for the potential prediction of response to ICI (Methods).

The orthogonal test subset was completely insulated from both the variant selection (TCGA STAD) and directionality (training set). When the selected variants were tested on the orthogonal test set ($n = 67$), the PRS was significantly increased in the responders (Wilcoxon rank sum test, $p$ value = 0.00071, Fig. 7A), and exhibited a higher area under the receiver operating characteristic curve (AUC, ROC) than PD-L1 expression (Fig. 7B), as well as against tumor mutational burden (TMB) or microsatellite instability (MSI) as calculated from WES data (Supplementary Fig. 7). Importantly, the information captured by the PRS score is a predictor independent of PD-L1 expression, and their combination, as well as potentially the integration of the expression or mutational status of additional genes, can be leveraged to further increase the prognostic accuracy of the model (Supplementary Fig. 8). The ilQTL PRS also shows higher generalizability in the test set as compared to standard PRS models generated with the top 3′-UTR or genic variants selected for their ability to predict response in the training set, showing the translational potential of 3′-UTR variants prioritized through the ilQTL analysis in a large discovery cohort (Supplementary Fig. 9). This proof-of-concept application showcases that non-coding variants can be used to predict immunotherapy treatment outcomes in cancer.

### Discussion
The increasing ease and lower cost of deep genome sequencing technologies will eventually allow the unbiased identification of non-coding variants with high confidence through the analysis of WGS data. However, the small number of currently available cancer samples with combined WGS and RNAseq data prohibits the use of WGS variant calling data for analyses that require high statistical power, such as QTL analysis. By repurposing RNAseq data available for hundreds of gastric adenocarcinoma patients in TCGA[30], we deeply investigated 3′-UTR somatic and germline variants in cancer. The use of RNAseq data allowed the identification of variants not only in the coding region of genes, but also in 5′UTR /3′-UTR sequences and non-coding genes like long non-coding RNAs (lncRNAs), which are omitted from standard WES assays, such as those performed for TCGA. To our knowledge, the

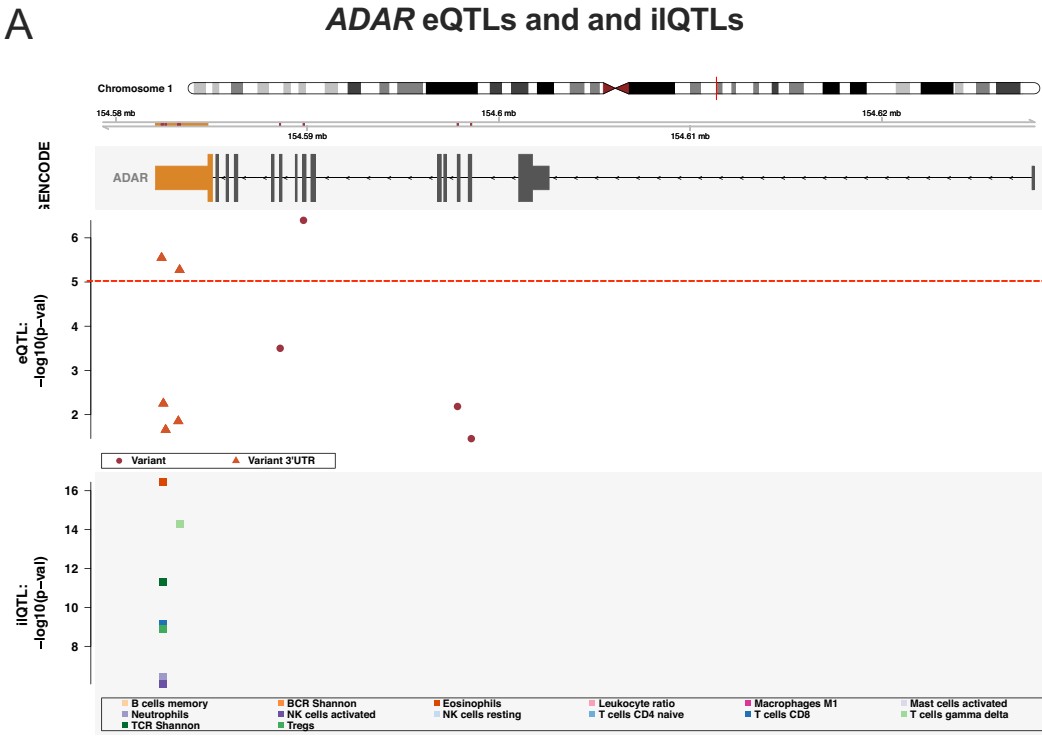

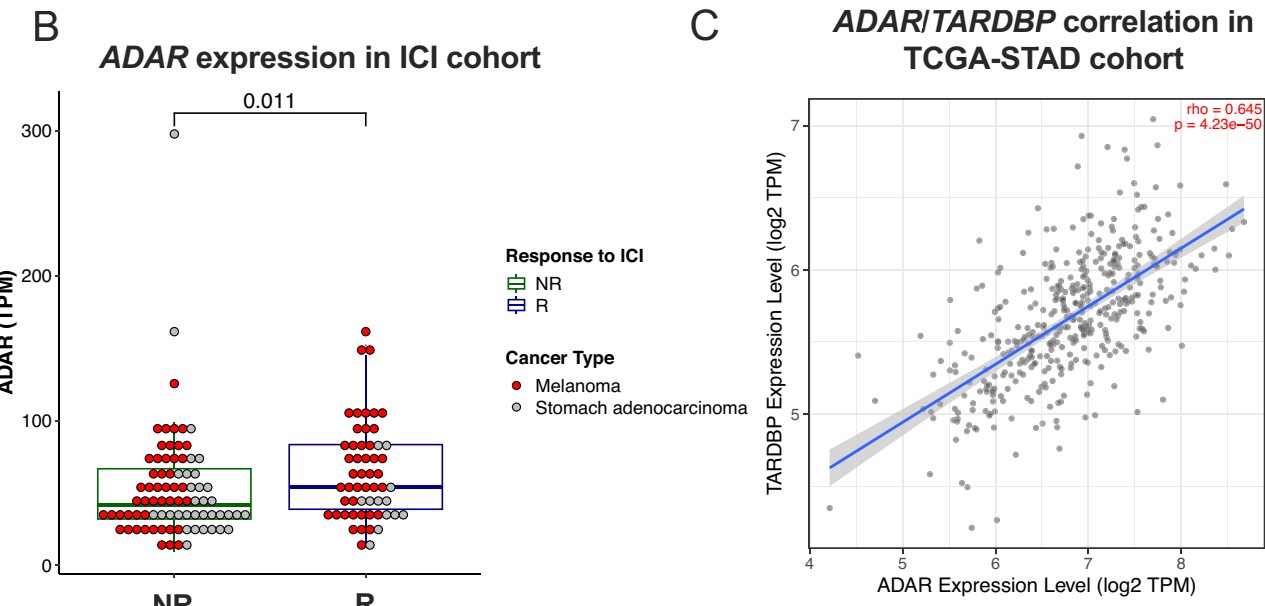

**Fig. 6 | Identification of causal 3′-UTR eQTLs and ilQTLs in *ADAR*. A** Plot showing the distribution of CDS and 3′-UTR eQTLs and ilQTLs for the *ADAR* gene. The y-axis represents the nominal eQTL *p* value for each variant. The exon where the 3′-UTR is found is colored orange. **B** A cohort of gastric cancer and melanoma patients was classified into responders (R, *n* = 80) vs non-responders (NR, *n* = 55) according to response efficacy with anti-PD-1 immunotherapy. Primary cancer RNAseq data from these patients were analyzed. Differential gene expression analysis revealed increased expression of *ADAR* in R compared to NR (two-sided Wilcoxon rank sum test, Benjamini Hochberg FDR correction). Boxplot lines represent the median and upper or lower quartiles, while upper whiskers represent the max and min. Gastric and Melanoma cancer types are colored with grey and red, respectively. **C** Correlation analysis between *ADAR* and *TARDBP* log2 TPM normalized expression performed using TIMER v2.0 (timer.cistrome.org). Data from 415 STAD patients are included and the Spearman's correlation coefficient (rho) and *p* value are reported. A linear regression line is shown in blue; the gray shaded area represents the standard error of the regression. Figure data are provided in the Source Data file.

only study that attempted to characterize 3′-UTR somatic variants from TCGA transcriptome-wide was by Wu et al.[32]. However, they mistakenly considered that the exome capture used in TCGA was the Illumina TruSeq Exome Enrichment Kit, which also targets 3′-UTRs, instead of the actual assays performed, where UTRs are not included and low coverage statistics are reported if the UTR regions are considered in the metrics[33]. Our analysis shows that 3′-UTR somatic variants are spread across the 3′-UTR regions as also supported by WGS assays (Fig. 1B) and not proximally to the CDS as previously reported[32], which was evidently due the TCGA WES not targeting 3′-UTR regions.

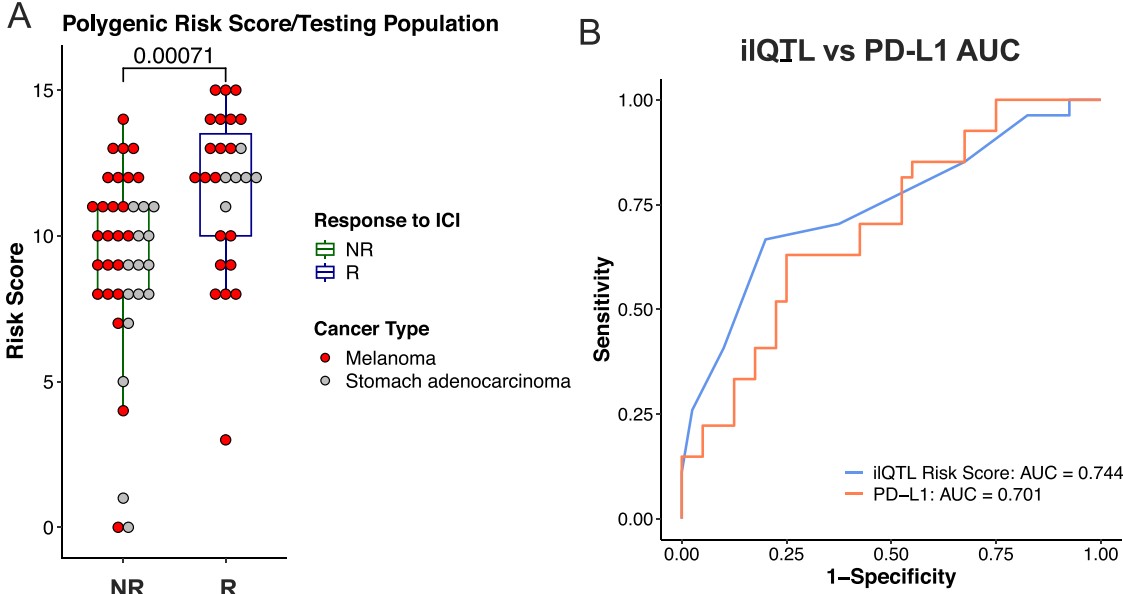

**Fig. 7 | ilQTL polygenic risk score for predicting response to immunotherapy in melanoma and gastric cancer patients. A** Comparison of the polygenic risk score (PRS) distribution in the non-responder (NR, *n* = 40) and responder (R, *n* = 27) groups of the testing population (two-sided, Wilcoxon rank sum test, Benjamini–Hochberg FDR correction). Boxplot lines represent the median and upper or lower quartiles, while upper whiskers represent the max and min.

Gastric and Melanoma cancer types are colored with grey and red, respectively. **B** A receiver operator characteristic (ROC) curve showing the ability of the PRS score and PD-L1 expression classifications to distinguish between R and NR patients in the testing population. An Area Under the Curve (AUC) score is reported for both classifiers. Figure data are provided in the Source Data file.

Despite the success of RNAseq data capturing variants in regions not covered by standard WES assays, the ability of this approach is confined to the expression space of the tumor of interest, since low (or no) expression directly inhibits variant calling. To address specificity concerns while maintaining high sensitivity, we followed GATK best practices for RNAseq data by deduplicating identical reads and by following through with variants identified in at least 1.3% of the population tested.

Our study reveals the importance of 3′-UTR variants in driving *cis*-gene expression in cancer and provides a framework for incorporating 3′-UTR variant-calling data in TCGA and other cohorts. We prioritized functional variants by performing a transcriptome-wide *cis*-eQTL analysis in the TCGA STAD cohort and identified significant variants across 1117 eGenes. 27.6% of the identified eQTLs overlapped with eQTLs identified from healthy stomach tissue from the GTEx consortium, with 90.6% of the overlapping significant variants being germline. This points not only to the difference between germline variants and somatic mutations but mostly to the regulatory divide between healthy and neoplastic tissue. A recent study comparing eQTLs generated with different combinations of germline/tumor variants and healthy/neoplastic tissue gene expression, concluded that the variation in the eQTLs they observed could be almost entirely attributed to the difference in the source material; highlighting further the genomic, epigenetic, transcriptomic, and regulatory differences observed between healthy and neoplastic tissue[73]. Around 90% of the 3′-UTR eQTLs overlap with putative or experimentally-supported miRNA and RBP binding sites, providing a potential functional relevance for those variants, with only 1.6% and 1.83% of somatic and germline variants colocalizing on the same microRNA and RBP binding site, respectively. The enrichment in immune-related pathways in the topmost significant 3′-UTR cis-eQTLs indicates the importance of 3′-UTR variants in controlling cancer immunogenicity. Since 3′-UTR regulatory roles go beyond post-transcriptional gene expression regulation and include localization, translation rate control, and even protein-protein interactions and liquid organelle formation[41],

we performed a transcriptome-wide ilQTL analysis. In this analysis as well, the majority of significant variants resided in 3′-UTR regions. This investigation shows that 3′-UTR variants can be associated with immune phenotype changes in an unbiased hypothesis-free manner. We discovered significant 3′-UTR ilQTL variants in widely studied immunoregulatory genes, such as *ADAR* and *STAT1*. One of the *ADAR* ilQTLs overlapped with multiple RNA binding sites, including TARDBP, an RNA binding protein and known regulator of ADAR expression in liver cancer and leukemia.

In addition to validating previously described 3′-UTR eQTLs, our approach also identified 3′-UTR variants and genes that have not been previously linked to immune-related functions in cancer. We utilized a massively parallel reporter assay (MPRA) to streamline validation across hundreds of candidate 3′UTR variants, with 15% exhibiting functional effects, even though the eQTLs were detected in patient samples and the MPRA assay was performed in gastric cancer cell lines, where the relevant RNA binding proteins, microRNAs and their targets might not exhibit conserved stoichiometry. The validation rate is comparable to the MPRA assay performed in Griesemer et al.[43]. Interestingly, among the significant hits from the MPRA assay, there were multiple genes encoding ribosomal subunits. Pathway analysis in the top significant CD8+ T cell infiltration QTL variants revealed an overall enrichment in ribosome-related proteins (Supplementary Fig. 4). Previous work has shown that changing the expression of ribosomal subunits can affect MHC class I presentation efficiency and the antigenic profile of a cell in the context of Influenza A virus infection, without altering translation efficiency[74], raising the question whether alteration of ribosomal proteins could have a similar phenotypic effect in cancer.

Finally, to investigate the clinical relevance of our ilQTL analysis, we showed that non-coding 3′-UTR ilQTL variants can predict response rates to immunotherapy treatments (Fig. 7A). In this study, we exploit the space of non-coding variants and show that a signature of ilQTL variants has stronger predictive power for drug response than PD-L1 expression in a cohort of melanoma and gastric cancer patients

(Fig. 7B). This is a direct application of non-coding mutations to predict response to immunotherapy, providing a potentially strong incentive to start including these important regulatory regions in WES investigations while the community waits to horizontally adopt WGS in somatic samples.

## Methods

### Cell culture

The AGS (ATCC CRL-1739) cell line was purchased from ATCC and the SNU-719 (KCLB-00719) cell line was purchased from the Korean Cell Line Bank (KCLB). AGS and SNU719 cells were maintained in RPMI-1640 (ThermoFisher Scientific, #11875093) with 10% FBS (ThermoFisher Scientific, A5256801). All cell lines were incubated at 37°C and 5% $CO_2$. The cell lines were verified by the vendors with STR profiling and they were tested for mycoplasma contamination at regular intervals using the MycoAlert Mycoplasma Detection Kit (Lonza, #LT07-318).

### RNAseq variant calling

Raw RNAseq data for 375 TCGA STAD primary cancer and 40 matched normal samples were obtained from Genomic Data Commons (GDC) following NIH dbGAP approval[75]. Reads were mapped against the human genome (hg38) using STAR[76]. Mapped reads were deduplicated and short variants/indels were called using Mutect2 following GATK (v4.1.4.0) best practices[34]. The Mutect2 output was converted to a gVCF format by using region coverage statistics. Since Mutect2 cannot perform genotype calling and does not distinguish between homozygous reference and no-call regions, HaplotypeCaller was also run in parallel by following GATK best practices for RNA[77]. For samples lacking a Mutect2 call at a specific variant position, HaplotypeCaller was used to distinguish whether the lack of a Mutect2 call was because of no coverage in that region or a homozygous reference genotype. A mutation was characterized as likely somatic by calculating the posterior probability of the event, while using variant call statistics, clonality in tumor samples, matched healthy tissues, and gnomAD variants[78] as priors. Calls were filtered using the FilterMutect2 tags "base_qual", "map_qual", "n_ratio" and "slippage". Only biallelic variants present in at least 5 of the 375 (>1.3%) samples were pursued further. Variants that were present in at least 2 out of the 40 matched normal samples comprised the Panel of Normals (PoN). Strelka2[36] was also run following the same preprocessing steps as for GATK callers.

### Germline variant calling

Affymetrix SNP array 6.0 data from blood samples for all TCGA STAD patients analyzed in this study were downloaded from GDC. The SNP array data were converted to a VCF format using birdseed2vcf (https://github.com/ding-lab/birdseed2vcf), and then whole-genome variant calls were imputed using Minimac4 and the 1000 Genomes Phase 3 project as a reference on the Michigan Imputation Server[79]. Somatic variants were further intersected against GTEx RNA editing events from REDIportal V2.0[35].

### eQTL analysis

Gene-level expression in TCGA STAD samples was calculated using Salmon v0.91 and Ensembl genome annotation v77[80,81]. A linear model was utilized to call eQTLs with FastQTL[39] following best practices[82]. In brief, gene expression across libraries was normalized using trimmed mean of m-values as implemented in edgeR[83]. Genes were selected based on an expression threshold of 1 read in at least 80% of the samples. An inverse quantile normal transformation was performed on the expression values prior to their inclusion into the linear model. Mutect2/HaplotypeCaller alternative allele dosage was utilized as genotype input, while age at diagnosis, sex and the top 5 genetic principal components (gPCs) and expression surrogate variables (SVs) were included as covariates[39,84], as follows: gene expression -Alt Dosage+age +sex+gPC1+gPC2+gPC3+gPC4+gPC5+SV1+SV2+SV3+SV4+SV5. Genetic

PCs were calculated using SmartPCA[85] on WES-derived germline variants from the same TCGA STAD patients, obtained from Huang et al.[86].

Following the nominal run of FastQTL[39], for each gene we selected variants that mapped in *cis* (within the genomic coordinates of the gene) and calculated a per-gene FDR-adjusted $q$ value for each variant. Genes that contained at least one variant with a $q$ value lower than 0.05 were defined as eGenes. A nominal threshold was defined for each eGene based on the highest nominal $p$ value that corresponded to a $q$ value < 0.05. The distribution of the threshold nominal $p$ value in all eGenes is shown in Supplementary Fig. 10. The median threshold nominal $p$ value (1e-7) was applied horizontally, across all eGenes, to describe significant calls (final cutoff: $q$ value < 0.05 and nominal $p$ value < 1e-7). eQTL calls were mapped to transcript annotations (Gencode v32) and relative genomic locations (5'UTR, CDS, 3'-UTR) were assigned using annotatr[87]. Only eQTLs with significant *cis* effects were retained for further analysis. Variant annotation for potential overlap with post-transcriptional regulatory regions was performed using the GenomicRanges package in R (v1.38.0)[88]. Experimentally-supported miRNA binding site coordinates were obtained from TarBase[47], predicted miRNA binding sites were acquired from microT-CDS[48], and CLIP-based predictions of RBP binding sites were obtained from the POSTAR2 database[49].

### ilQTL analysis

Immune data per sample were obtained from the CRI iAtlas project[57]. QTL analysis was performed with FastQTL, using the same genotype and covariate data as above, while using quantile-normalized immune profile estimates as phenotypes. ilQTL selection as well as genomic and regulatory annotation were performed as for eQTLs.

### Enrichment and over-representation analyses

For pathway enrichment/over-representation analyses, the top 500 eGenes, ranked based on their lowest nominal $p$ value, were investigated by pathway enrichment analysis using ClusterProfiler (v3.12.0)[89]. Pathway information was obtained from the Gene Ontology Resource[90] and the Kyoto Encyclopedia of Genes and Genomes (KEGG) Pathway database[91]. Plots were generated in R using ggplot2 (v3.3.3). One-sided Fisher's exact test was utilized to evaluate cell type-specific enrichment of eQTL genes against the reference marker gene sets from the gastric cancer single-cell atlas established by Sun et al.[46], corresponding to individual cell types or subtypes.

Locus permutation analyses were performed with RegioneR[92] utilizing a resampling ($n = 5000$) permutation test. As a sampling space, all 3'UTR regions were split into 50 bp segments using a walking window of step = 1 ($n = 39,142,309$ windows). All evaluated regulatory regions from microT, TarBase, and POSTAR were included in the sampling space.

### Cancer immunology-related genes

A collection of more than 2500 immune-related genes was manually curated from the literature and experimental resources (Supplementary Data 2). Specifically, the list includes immune checkpoint and immunomodulatory genes, genes involved in the MHC machinery and microsatellite instability, cytokines and chemokines[93–95], gene markers for metabolic reprogramming[93,96] and oncogenes or complexes that can affect the tumor transcriptional and immune landscape, such as EZH2−PRC2 chromatin remodeling complex members and BAF/PBAF complex members[97,98]. We also incorporated significant genes from hypothesis-free CRISPR−Cas9 screens for CD8+ T-Cell effector function[54,55] and in vivo screening of transplantable tumors in mice treated with immunotherapy[56]. In addition, we included the Urea cycle (GO:0000050) and Mismatch repair (GO:0006298) Gene Ontology terms, the list of Human DNA repair genes from Lange et al.[99], and the following entries from the Kyoto Encyclopedia of Genes and Genomes

(KEGG): MAPK signaling pathway (hsa04010), PI3K-Akt signaling pathway (hsa04151), Wnt signaling pathway (hsa04310), JAK-STAT signaling pathway (hsa04630), Antigen processing and presentation (hsa04612). Finally, the list comprises more than 250 genes from signatures associated with response to ICI[50–53].

## MPRA pool design

A MPRA was performed with eQTL variants in eGenes included within the curated cancer-immunology gene list that had a nominal *p*-value lower than 1e-5 (Supplementary Data 3). The nominal *p* value threshold was higher than that for other downstream eQTL analyses to allow the capture of causative effects by less common variants, such as rare somatic eQTLs. A pool of 19220 150 bp-long oligonucleotides (oligos) was synthesized commercially (Twist Biosciences). Each oligo contained the reference or alternative allele of the eQTL variant in the middle flanked by 50 bp of reference transcriptomic sequence on either side. In the case of eQTLs residing in alternative transcript isoforms, oligos were synthesized for all possible transcripts. For eQTLs that were close to the end of the transcript a random sequence was added to bring up the length of the sequence to 101 bp. The random sequence was the same in all oligos and was selected to lack any predicted 7 or 8 bp-long RBP and miRNA-seed binding sites.

An 8 bp-long barcode was added at the 3′ end of the 101 bp-long sequence. Each allele was represented by 10 unique barcodes. All 8 bp barcodes that matched RBP or miRNA seed binding sites were removed[100,101]. Finally, 20 bp sequences were added on either side of the oligo that matched the 5′ and 3′-end sequences of the XhoI-/NotI-digested psicheck2 vector (Promega, C8021) to allow cloning with the NEBuilder HiFi DNA Assembly kit (NEB, E2621S).

## Pool amplification and cloning

Amplification of the pool, prior to cloning, was performed using 0.5 µM of each of the PCR_lib_fwd and PCR_lib_rev primer pair (Supplementary Data 5) with the NEB Next High-Fidelity 2x PCR Master Mix (NEB, M0541L). The following PCR conditions were used: 98 °C for 30 sec, 20 cycles (98 °C for 10 sec, 63 °C for 10 sec, 72 °C for 15 sec), 72 °C for 2 min. The amplified oligo pool was introduced into a XhoI-/NotI-digested psicheck2 vector using NEBuilder HiFi DNA Assembly kit (NEB, E2621S), as per the manufacturer's protocol. The assembly reaction product was purified following a standard isopropanol precipitation protocol, as described in Joung et al.[102]. The purified plasmid pool was transformed into Endura ElectroCompetent cells (Lucigen, #60242-1) at 50 ng plasmid per 25 µl of bacteria ratio, following the provider's protocol. A total of 8 transformation reactions were pooled together and plated onto large 15 cm LB Agar plates at 37 °C for 12 h. A large enough number of colonies to ensure at least 500 colonies/oligo representation was harvested directly from the LB Agar plates and the plasmid pool was purified by performing at least 2 midipreps per 15 cm LB Agar plate, using the Qiagen Plasmid Plus Midi kit (Qiagen, #12943).

## MPRA transfection and library prep

SNU719 and AGS cells were seeded in 15 cm plates to achieve 80% confluence the next day. Cells were transfected with 10 µg of the MPRA plasmid library using TransIT-X2 reagent (Mirus Bio, MIR 6004) as per the manufacturer's protocol, aiming for a transfection efficiency of 50-80%. Total RNA was collected 48 hr post-transfection using the miR-Neasy mini kit (Qiagen, #217004). Genomic DNA was removed using the Turbo DNA-free kit (ThermoFisher Scientific, AM1907) following the manufacturer's "Rigorous DNase treatment" protocol. Per replicate, 15 µg total RNA was reverse transcribed with SuperScript IV Reverse Transcriptase (ThermoFisher Scientific, #18090010) using oligo-dT primers. Amplicon sequencing libraries from cDNA or plasmid pool DNA were constructed through two PCR reactions, adapted from Pinto et al.[103]. In the first PCR round, 1:10 diluted cDNA was amplified using 0.2 µM of the DT_barcodePE_Fv2 and 0.2 µM of an equimolar mix of the DT_barcodePE_Rv2 primers (0 to 6 random Ns, Supplementary Data 5). The PCR reaction was performed with the NEB Next High-Fidelity 2x PCR Master Mix (NEB, M0541L) and the following conditions: 98 °C for 30 sec, 10 cycles (98 °C for 10 sec, 63 °C for 10 sec, 72 °C for 15 sec), 72 °C for 2 min. Enough PCR reactions were run to ensure that all the cDNA from each replicate was amplified. In the second PCR reaction, 1:10 diluted PCR round 1 product was amplified using 0.5 µM of a unique pair of multiplexing Illumina primers (PE_i5 and PE_index in Supplementary Data 5). The following PCR conditions were used: 98 °C for 30 sec, 10 cycles (98 °C for 10 sec, 62 °C for 10 sec, 72 °C for 15 sec), 72 °C for 2 min. For each replicate, the second round PCR product was purified through gel extraction using the Monarch Gel Extraction kit (NEB, T1020S). The quality of each library was assessed by an Agilent Tapestation D1000 assay (Agilent). An equimolar mix of all libraries was sent for single-end 150 bp sequencing on an Illumina sequencer, with 20% PhiX spike-in to increase library complexity. The mixed library was sequenced at a depth to ensure at least 10 M reads per replicate (>500 reads per oligo).

## MPRA analysis

MPRA analysis was performed similarly to Mattioli et al.[104]. Briefly, barcode counts were calculated from raw reads and then normalized per sample based on sequencing depth. For each sample, a barcode RNA to DNA ratio was calculated by dividing the barcode counts in each replicate to that in the plasmid pool library. The RNA to DNA ratios were then log-transformed and quantile normalized across samples. A two-sided Wilcoxon test was performed to compare barcode count ratios between reference and alternative allele oligos in each replicate. To combine replicate p-values, the Stouffer's method was used, and Benjamini-Hochberg false discovery rate (FDR) correction was applied. Each oligo was represented by 10 barcodes, so to obtain a per-oligo activity in each sample, the median activity was calculated. Fold-change was defined as the ratio of the alternative to the reference allele median activity.

## ICI cohort analysis

Pre-treatment tumor RNAseq data were retrieved from four published ICI studies (anti-PD1 or anti-CTLA4 treatment), of which three addressed melanoma patients (*n* = 90)[50,51,69] and one addressed gastric cancer patients (*n* = 45)[70]. The combined cohort (*n* = 135) included 55 responders and 80 non-responders to immunotherapy. Gene expression of pre-treatment tumor samples was quantified from RNAseq reads using Salmon v0.91[80]. To calculate differential expression of *ADAR* between responders and non-responders, a Wilcoxon rank sum test with continuity correction was performed. The ICI cohort was also randomly split into a training (*n* = 68) and a testing set (*n* = 67) and the Variant Calling and ilQTL pipelines were run on the pre-treatment tumor RNAseq data. The 3′-UTR ilQTL variants enriched in ICI responders in the training set (*n* = 28, FDR-adjusted *p* value < 0.05) were selected to comprise the PRS. The score is calculated as the number of variants detected in the patient's tumor sample, therefore ranging from 0 to 28. All variants were present across the cohort in 1 or more individuals, with 15 being present in the smaller gastric cancer sub-population (*n* = 45).

PD-L1 expression and the ilQTL signature were combined with a multivariate linear model, and its performance was assessed in the test set (*n* = 67). For the PRS calculation of the 3′-UTR and CDS + UTR regions, a similar approach was followed as described above. The top 28 enriched variants based on the odds ratio per gene in 3′-UTR or CDS + UTR in the R vs NR samples of the training set were selected accordingly (one-sided Fisher's exact test *q* value < 0.05). Only the most highly enriched variant per gene was included in the final models. All PRS models (ilQTL, ilQTL + PD-L1, 3′-UTR, Genic Variants) were trained and tested on identical patient sets.

## Whole exome sequencing analysis

WES data for melanoma[51] and gastric cancer[70] were downloaded from SRA using the sra-toolkit and processed according to GATK best practices using GATK 4.4[105]. Briefly, FASTQ files were checked for presence of contamination using FastQC 0.12.1[106] and MultiQC 1.17[107], and following inspection, were aligned using the Burrows-Wheeler Aligner (BWA) 0.7.17[108] using the BWA-MEM algorithm against the hg38 genome distributed by the GATK team (https://console.cloud.google.com/storage/browser/genomics-public-data/resources/broad/hg38/v0). The resulting SAM files were sorted and indexed using samtools 1.18[109]. The files were then post-processed, marking duplicates and running Base Quality Score Recalibration (BQSR). A panel of normals was generated for each study using the healthy patient samples, which was then used along with each tumor-normal WES pair to call mutations using Mutect2, with gnomAD as a germline resource. To minimize artifact calls and contamination, the read orientation artifact workflow was followed before filtering the Mutect2 calls. To accelerate runtime, intervals were used where available, utilizing the capture kit information for each study. SnpEff v5.2[110], with the GRCh38.105 database, was used to annotate the resulting VCF files, and then the TMB was calculated with pyTMB v1.3[111], using a variant allele fraction of 0.05, a MAF of 0.001, minimum depth of 20 and minimum alternative depth of 2 to minimize noise, while filtering out low quality, non-coding, synonymous and polymorphic calls against gnomAD[78]. For the MSI calculation, MSIsensor-pro v1.2.0[112] was used by first scanning the reference genome for microsatellite information and then running in tumor-normal mode on the aligned reads with default parameters.

## Statistics and reproducibility

All statistical analyses were performed using R or GraphPad Prism 8. The QTL analysis was performed with 375 STAD patients for which TCGA contained primary cancer RNAseq data. The MPRA assay (Fig. 4A, B and Supplementary Figs. 3A, B and 6A, B) was independently repeated three times in two gastric cancer cell lines. No statistical method was used to predetermine sample size. No data were excluded from the analyses. Randomization was utilized in the selection of the training/testing sets for the predictive models as described in the Methods section. The Investigators were not blinded to allocation during experiments and outcome assessment.

## Reporting summary

Further information on research design is available in the Nature Portfolio Reporting Summary linked to this article.

## Data availability

The data that support the findings of this study are derived from publicly available datasets. Raw RNAseq, WES, and SNP array data for gastric adenocarcinoma patients were obtained from The Cancer Genome Atlas (TCGA) database at gdc.cancer.gov. The TCGA barcodes of the STAD patients included in the study are provided in the Supplementary Information. The PRS and predictive model analyses were performed on a combined cohort of publically available melanoma and gastric cancer pre-treatment RNAseq data obtained from the Gene Expression Omnibus (GEO) database under accession codes GSE115821[50], GSE78220[51], and GSE91061[69], and the European Nucleotide Archive (ENA) under accession PRJEB25780[70]. Matched WES data for patients included in the PRS and predictive model analyses were obtained from the Sequence Read Archive (SRA, https://www.ncbi.nlm.nih.gov/sra) under accessions SRP067938 and SRP090294[51] and from ENA under accession ERP107734[70]. The raw MPRA amplicon sequencing data generated in this study have been deposited in the GEO database under accession code GSE261709. The remaining data are available within the Article, Supplementary Information, or Source Data file. Source data are provided with this paper.

## Code availability

Computational analyses are described in detail in the methods section. The majority of analyses have been performed utilizing publicly available tools. Custom code utilized for the MPRA analysis has been deposited at https://github.com/ivlachos/3UTR.

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

## Acknowledgements

We acknowledge the NCI award RO1CA258776 to ISV and the NCI Outstanding Investigator Award R35CA232105 to FJS. The results presented here are based upon data generated by the TCGA Research Network: https://www.cancer.gov/tcga. TCGA data were accessed through dbGaP phs000178.v11.p8. Portions of this research were conducted on the Ithaca High-Performance Computing system, Department of

Pathology, BIDMC, and the O2 High-Performance Compute Cluster at Harvard Medical School.

## Author contributions
I.S.V., F.J.S., and C.M. conceived and designed the study. C.M. performed the variant calling, e/ilQTL analysis, and MPRA assay and analysis. Y.M. performed the single-cell enrichment analysis and contributed to the 3'UTR variant analyses; X.L.K. and F.B. performed functional enrichment analyses. E.K. and D.K. developed and applied the PRS and predictive models. C.M. and D.K. designed the MPRA pool. K.M. contributed to the MPRA assay analysis. S.N. analyzed the ADAR expression in ICI response cohorts. N.K. performed the WES analysis. Y.P.J. and V.R.R. reviewed the data and provided significant intellectual input. I.S.V., F.J.S., and C.M. wrote the manuscript with input from all authors. I.S.V. and F.J.S. provided oversight for the study.

## Competing interests
I.S.V. consults for Guidepoint Global, Cowen, Mosaic, and NextRNA. Beth Israel Deaconess Medical Center has filed a patent application based on this work for "Methods and compositions related to non-coding variants for the prediction of response to cancer immunotherapy" under 63/378,392, where I.S.V., F.J.S., C.M., and E.K. are named as co-inventors. The remaining authors declare no competing interests.
