## [Peer Review File · Nature Communications]

Determinants of gastric cancer immune escape identified from non-coding immune-landscape quantitative trait lociEditorial Note: This manuscript has been previously reviewed at another journal that is not operating a transparent peer review scheme. This document only contains reviewer comments and rebuttal letters for versions considered at Nature Communications.

Reviewers' Comments:

Reviewer #1:

Remarks to the Author:

I previously reviewed this study when it was submitted to another journal. As such, some of my comments apply to the Author's responses to my previous comments, and other comments are based on the current submitted version.

Major Comments

1) The author's response to my 1st comment is not satisfactory. All I am requesting is a clear demarcation of the variants – what percentage are germline, what percentage are somatic (caused by DNA mutations), and what percentage are due to RNA-editing. This is not provided. In addition, the authors need to clearly state to what degree the associations they are seeing (cis effects on expression, immune landscape phenotypes, etc) are driven by germline vs somatic (DNA) vs somatic (RNA editing)

2) The response from the authors "Specifically, the majority of our findings (77%) are not identified in healthy stomach tissue (GTEx), showcasing the differences between germline and somatic mutations and the regulatory divide between healthy and neoplastic tissue" conflates two very different issues. Are the differences between normal and gastric cancer due to either germline vs somatic variants, or between normal and gastric cancer expression states. It is entirely possible that the same germline variant will show a eQTL effect in one cellular state and not another. If the key novelty (as the authors proclaim) of the paper is the somatic variants, then the authors should be clear which effects are due to somatic variants.

3) The authors statement that there are only a handful of matched gastric cancer WGS + RNA-seq data sets is probably true, but that does not excuse them from mining the plentiful gastric cancer WGS data (alone) to see if additional variants are germline (being seen in multiple cases).

4) Is the polygenic risk score (PRS) for predicting immune checkpoint response superior if one uses a PRS based on 3' UTRs vs the entire genome which would be the 'standard approach'?

5) Please be very clear in the Results the amount of genomic region (in Mb or Kb) corresponding to 3' UTRs that are *not* found in the TCGA WES kits. This is relevant as it may motivate other investigators to perform similar analyses for other cancer types

6) Figure 1B is highly dubious. Why is there a proximal accumulation of variants in the ICGC WGS, while there is an even distribution on the RNA-seq? This could be explained (perhaps) by the ICGC WGS data set being somatic (DNA) calls, while the RNA-seq calls are a combination of germline and somatic (DNA) + somatic (RNA editing). However, if this is the case, then this would logically indicate that the RNA-seq calls are largely dominated by germline calls given the uniformity. If the nature of the variant calls is suspect even in Figure 1, the entire paper is called into question.

7) It should be acknowledged in the Abstract that 60% of the variants are likely to be germline and of

the remaining almost half (38-40%) are likely due to RNA editing, and that these are *lower limits*. The percentage of germline variants will likely increase if one further intersects those that have rs IDs. As a sanity check, the final set of variants that are "non-germline" and "non-RNA editing" should follow a curve similar to the ICGC WGS calls.

8) The three PD-L1 3' UTR variants tested should be referred to as gene polymorphisms, not mutations. Please use this nomenclature throughout the manuscripts (polymorphisms refer to germline variants).

9) For the subset of variants that are present in "5 or more samples, corresponding to 1.3% or higher minor allele frequency (MAF) in the tested population", what percentage falls into the germline category (see comment #5) and which somatic (DNA + RNA-editing)? I suspect that these will be enriched significantly above and beyond the 60% of germline variants. Here, my understanding is that the 60% figure highlighted in lines 320-322 is *before* the 5 or more samples threshold.

8) Somatic-only eQTLs are of interest (line 358-360) – can the authors highlight some examples?

9) How do the prioritized variants tested in the reporter assay map across the categories of germline/somatic(DNA)/somatic(RNA editing)?

10) Is the association of variants with immune phenotypes (iQTLs) specific to immune phenotypes? How about other cancer hallmarks such as signatures of cell proliferation etc? The point that iQTLs appear to be somatic (lines 418-419) is of great interest – can the authors point to a few examples and show the evidence confirming the somatic nature?

11) It should be acknowledged clearly (I would recommend at line 451) that the association of ADAR1 overexpression with ICI response is counter to the Manguso et al findings. This discrepancy should be discussed (in the Discussion). If the authors want to be 'provocative' (their own words), then they should be upfront about it.

12) Some functional validation would be useful – for example, a knockdown of TARDBP and subsequent measurement of ADAR1 levels. Given the institutes the authors hail from, this should be absolutely trivial.

13) Can the authors test if the combination of iQTL PRS + PD-L1 improves ICI response prediction? Also is the power conferred by iQTL independent of PD-L1? This should be tested by multivariate analysis.

Minor Comments

1) It should be acknowledged in the Introduction that RNA editing has a preference for UTR regions (PMID 22448268 and associated manuscripts)

2) Some sentences are difficult to understand : "In PCAWG, RNAseq data analysis identified driver alterations in all 87 samples without a driver mutation at the DNA level" – all 87 samples out of how many samples? Without such details, it is not possible to evaluate such claims

3) Shouldn't Figure 3C be normalized by genomic region length on the y-axis?

4) How many iQTLs were identified? This number is not provided in the Results (line 413)

Reviewer #5:

Remarks to the Author:

Dear Editor,

After carefully reviewing the manuscript and the author's response to Reviewer #3 and Reviewer #4, my overall judgment is that the authors have mostly responded well to the questions. However, in some reasonable questions, particularly Reviewer #3: Major: 8, 9, 10, and Minor: 8, and Reviewer #4: Major 3, specific comments: Page 25, the authors did not make any substantial changes in the manuscript and their explanations were not very clear. Therefore, my suggestion is that the manuscript should be considered for publication only if the necessary revisions requested by Reviewer #3 and #4 are made. The following are my comments on the author's response to Reviewer #3 and #4.

Reviewer #3:

Major:

Question 1: The authors responded well to this question. They clarified the discrepancy observed by the reviewer and explained that the 5,431,118 variants they identified are spread across the genome.

Question 2: The author provided a reasonable and acceptable response by pointing out that the lower number of eGenes found compared to some other studies is due to the heterogeneity in Calabrese et al.'s (2020) nature.

Question 3: The author correctly pointed out that the Griesemer study only utilizes germline variants and not somatic 3' UTR mutations. Therefore, as they claimed, it is indeed the first study to utilize a massively parallel reporter assay to evaluate somatic 3' UTR mutations.

Question 4: The author has now provided the permutation test results in the manuscript, which is necessary as suggested by the reviewer.

Question 5: Reviewer's question 5 is somewhat confusing, and the author has made their best effort to answer it. I believe their response is satisfactory.

Question 6: As the reviewer pointed out, the validation rate is not very high. However, considering the complexity of the experimental conditions and the cell lines used, it is understandable. Therefore, I think the authors have tried their best to answer this question and have done so properly.

Question 7: The authors have now rephrased the sentence in the manuscript for better understanding, which is acceptable.

Question 8: Reviewer #3 simply wants to know more about the TARDBP binding motif, and I think it is reasonable to ask this question. The author should at least discuss it in the discussion section.

Question 9: Reviewer #3 requested the authors to plot the results separately for both the melanoma and gastric adenocarcinoma cohorts. Although the authors plotted the results of gastric cancer in the response, they did not make any changes in the manuscript. I agree with Reviewer #3 that they should provide individual plots for these two cohorts in Figure 7a and Figure 6b. Alternatively, they should color the dots in Figure 6b and 7a to differentiate these two cancer types so that readers can understand that the difference between R and NR is independent of cancer type and not confounded by cancer types.

Question 10: The reviewer pointed out if the authors could compute the QTLs directly for R and NR as the phenotype in the training set and evaluate them in the test set. I think this is a good suggestion for the authors to try, and they should add the results at least in the supplemental material to compare with the results in Figure 7b. It is a good and not that complex approach for the authors to further support their claim that "this proof-of-concept is the first time that non-coding variants have been used to predict immunotherapy treatment outcomes in cancer."

Minor:

Question 8: The reviewer pointed out that "ADAR appeared as a hotspot for 3' UTR eQTL...." This statement of "hotspot" should be supported with statistics, which I agree with. The authors should revise their claim in the manuscript accordingly.

Reviewer #4:

Major:

Question 1: The authors' response to this question was partially satisfactory. While I agree that trans effect analysis is beyond the scope of the present study, I believe they should at least discuss the possibility of a convergent function or mechanism of germline and somatic mutations in the 3' UTR of the ADAR gene, as suggested by the reviewer. This discussion would enhance the comprehensiveness of the manuscript.

Question 2: The authors' response regarding the MPRA assay is reasonable. It is indeed a valuable approach to validate in silico predictions in a high-throughput manner. However, it is important to acknowledge that the validation rate is limited, which is also mentioned by Reviewer #3. The authors should address this limitation at least in the discussion.

Question 3: I agree with Reviewer #4's suggestion to compare the performance of the proposed biomarkers with other well-known biomarkers such as tumor mutation burden and MSI. The references (77) for gastric cancer and (68) for melanoma contain the necessary data. For example, ref 77 dataset is available here: <https://www.ebi.ac.uk/ena/browser/view/PRJEB25780>. "I am confused by the authors' claim that they are unable to obtain these metrics for comparison. They should be able to quickly compare the AUC from these data and include the results, at least in the supplemental material. Therefore, I strongly recommend that the authors perform the analysis as suggested by Reviewer #4.

Specific comments:

Page 25: The reviewer questioned how well these 28 genes are represented in the gastric cancer dataset. This is an important point to address, and the authors responded by stating that 15 out of the 28 variants were found in at least one sample. I believe this is crucial information that should be included in the main text of the manuscript.

Other minor questions that I believe should be addressed:

1. In Figure 5, the figure legend should mention a dark line instead of a dashed red line.
2. In Figure 3a, the author only provides a very rough formula for eQTL and doesn't even include the genotype in the formula. I suggest they provide a detailed formula in the methods section.

I hope these revisions and clarifications will strengthen the manuscript. Thank you for considering my comments.

Sincerely,
Qiyuan Li

We would like to thank the Reviewers for their insightful comments. We believe that they have significantly contributed to improving our manuscript. Following the Reviewers' recommendations, the revised version provides additional evidence and incorporates helpful clarifications.

Please find our detailed responses to each comment below.

REVIEWER COMMENTS

Reviewer #1 (Remarks to the Author):

I previously reviewed this study when it was submitted to another journal. As such, some of my comments apply to the Author's responses to my previous comments, and other comments are based on the current submitted version.

We would like to thank the Reviewer for their comments.

Major Comments

1) The author's response to my 1st comment is not satisfactory. All I am requesting is a clear demarcation of the variants – what percentage are germline, what percentage are somatic (caused by DNA mutations), and what percentage are due to RNA-editing. This is not provided. In addition, the authors need to clearly state to what degree the associations they are seeing (cis effects on expression, immune landscape phenotypes, etc) are driven by germline vs somatic (DNA) vs somatic (RNA editing)

We have edited the text to report not only percentages, as was previously performed, but also the exact numbers of germline, somatic, and probable editing sites, wherever variants or relevant results are reported.

Specifically, the text now reads:

Section reporting all variants

"Out of 5,431,118 variants identified across the genome by the RNAseq GATK analysis post-filtering, 3,283,340 (60.5%) overlapped with germline variant calls identified from blood samples from the same patients. The remaining 2,147,778 variants (39.5%) were treated as "likely somatic" calls. Of the likely somatic calls, 1,429,039 variants (66.5%) intersected with common RNA editing events identified in the GTEx database"

Section reporting the variants incorporated in the eQTL/iQTL analysis

"We investigated variants that were present in 5 or more samples, corresponding to 1.3% or higher minor allele frequency (MAF) in the tested population (2,917,776 total variants, 68,4% / 1,994,516 germline, 31.6% / 926,260 somatic, in which 67.2% / 620,419 overlapping with editing sites"

Section reporting eQTLs

"With a cutoff of a nominal p-value of $1e-7$, we identified ~3,000 cis-eQTLs in protein-coding genes, accounting for 75% of all eQTLs (Fig. 3B). Out of the 3,133 exonic (CDS/UTR) variants in protein coding genes, 1,845 / 58.9% were germline and 1,288 / 41.1% were somatic (254 overlapping with editing sites)."

Section reporting iQTLs

“A total of 1,715 ilQTLs were identified, with 159 (9.3%) identified as germline. Among the remaining 90.7% / 1,556 “likely somatic” ilQTLs, 370 intersected with common RNA editing events.”

2) The response from the authors “Specifically, the majority of our findings (77%) are not identified in healthy stomach tissue (GTEx), showcasing the differences between germline and somatic mutations and the regulatory divide between healthy and neoplastic tissue” conflates two very different issues. Are the differences between normal and gastric cancer due to either germline vs somatic variants, or between normal and gastric cancer expression states. It is entirely possible that the same germline variant will show a eQTL effect in one cellular state and not another. If the key novelty (as the authors proclaim) of the paper is the somatic variants, then the authors should be clear which effects are due to somatic variants.

We would like to thank the Reviewer for the comment. As mentioned in the previous comment, we have provided clarifications for the somatic/germline origin of all variant calls and significant regulatory events.

Regarding the % overlap with healthy gastric tissue eQTLs, we agree that it is most probably a combination of germline/somatic but also due to the regulatory divide between healthy and neoplastic tissue.

We have added a clarification of the number and percentage of the overlapping germline/somatic eQTLs between tumor and healthy gastric tissue, as well as a more detailed explanation of what could be the underlying mechanisms in the discussion section.

We reference also the recent study of Sheng and colleagues (Briefings in Bioinformatics, 2020) who performed an interesting exercise comparing eQTLs with variants/mutations from germline vs tumor, and gene expression from healthy vs tumor tissue. They conclude:

“Considering that the analyses were performed using the exact same set of SNPs and genes, the variation in the eQTLs can be almost entirely attributed to the difference in the source material (blood, normal tissue and tumor tissue).”

The relevant text in the results now includes that 90.6% of the overlapping eQTLs with GTEx are germline

Discussion section

“... eQTLs overlapped with eQTLs identified from healthy stomach tissue from the GTEx consortium, with 90.6% of the overlapping significant variants being germline. This points not only to the difference between germline variants and somatic mutations but mostly to the regulatory divide between healthy and neoplastic tissue. A recent study comparing eQTLs generated with different combinations of germline/tumor variants and healthy/neoplastic tissue gene expression concluded that the variation in the eQTLs can be almost entirely attributed to the difference in the source material; highlighting further the genomic, epigenetic, transcriptomic, and regulatory differences observed between healthy and neoplastic tissue...”

3) The authors statement that there are only a handful of matched gastric cancer WGS + RNA-seq data sets is probably true, but that does not excuse them from mining the plentiful gastric cancer WGS data (alone) to see if additional variants are germline (being seen in multiple cases).

We would like to thank the Reviewer for the recommendation. We believe that having matched germline data, derived from blood samples of the same subjects, is a more accurate resource for the characterization of the somatic/germline origin as compared to data derived from different subjects and/or populations. We believe that the addition of matched germline data for all 375 subjects of the cohort provides an accurate representation of the somatic vs germline percentage presented above.

4) Is the polygenic risk score (PRS) for predicting immune checkpoint response superior if one uses a PRS based on 3' UTRs vs the entire genome which would be the 'standard approach'?

We would like to thank the Reviewer for this comment. We believe that it's important to clarify that the point of the PRS investigation was not to derive a score tailored to this specific cohort but to show that functional 3'UTR variants affecting the tumor microenvironment (ilQTLs), identified in an orthogonal population (TCGA), can be directly applied to other cohorts and enable patient stratification with high accuracy, while reducing the need for a large training population.

This approach stresses clearly the point that these 3'UTR variants (ilQTLs) are not only functional and important but can have a direct translational impact.

However, we agree with the Reviewer that evaluating a simple approach tailored to this cohort could also provide useful information.

Following the Reviewer's suggestion, we calculated the Polygenic Risk Score directly, without using the ilQTL annotation, and incorporated all 3' UTR variants and all variants on CDS and UTR regions identified in the screening. As in the ilQTL PRS, the samples were separated into a training (n=68) and a test (n=67) set, and the top 28 enriched variants based on the odds ratio in the R vs NR samples of the training set were selected. The number of variants (n=28) as well as the training and test set were identical to those used for the ilQTL PRS model. The variants were then utilized to devise a polygenic risk score for the potential prediction of response to immune checkpoint inhibition.

As seen in the figure, the ilQTL PRS generalizes better in the test set as compared to the 3' UTR and All Variant models, majorly due to the lack of power to generate a robust PRS model with just the 68 training cases. This is not the case for the ilQTLs, since they represent biological and regulatory mechanisms.

The requested analyses provide further evidence for: a) that the ilQTL analysis is an effective tool for the prioritization of functional 3'UTR variants with translational potential, and b) the prioritized variants and models have cross-study applicability, as per our original hypothesis. Both aspects strengthen the importance of 3'UTR variant detection and investigations in cancer, which is an aspect missing in a large number of studies.

The plot below has been added as Supplementary Figure S10 in the manuscript, as well as the text below:

Fig. S10: Comparison of ilQTL 3' UTR PRS with 3' UTR and genome-wide PRS in distinguishing responders and non-responders to ICI. Receiver Operator Characteristic (ROC) curves showing the ability of the PRS score classifications, calculated based on significantly top enriched variants from (i) ilQTL (n=28, blue), (ii) 3' UTR (n=28, orange), (iii) all enriched variants in CDS and UTR regions (n=28, cyan), coupled with PD-L1 expression classification to distinguish between R and NR patients in the testing population (n=67). An Area Under the Curve (AUC) score is reported for all models.

And the relevant text has also been added:

“The ilQTL PRS also shows higher generalizability in the test set as compared to standard PRS models generated with the top 3'UTR or genic variants selected for their ability to predict response in the training set, showing the translational potential of 3'UTR variants prioritized through the ilQTL analysis in a large discovery cohort (Fig. S10)”

Methods Section

“For the PRS calculation of the 3' UTR and CDS + UTR regions, a similar approach was followed as described above. The top 28 enriched variants based on the odds ratio per gene in 3' UTR and genome-wide in the R vs NR samples of the training set were selected accordingly (one-sided Fisher's exact test q -value < 0.05). Only one (top eQTL) variant per gene was included in the final models. All PRS models (ilQTL, ilQTL + PD-L1, 3'UTR, All Variants) were trained and tested on identical patient sets”.

*5) Please be very clear in the Results the amount of genomic region (in Mb or Kb) corresponding to 3' UTRs that are *not* found in the TCGA WES kits. This is relevant as it may motivate other investigators to perform similar analyses for other cancer types*

We agree with the Reviewer that the lack of coverage of 3'UTR regions is indeed crucial and by stressing it further, it could motivate the community to perform additional analyses. The complete lack of 3'UTR regions

from TCGA or from all WES kits has been repeatedly mentioned in the literature, even contemporaneously with the TCGA studies (Sulonen, *et al.* Genome Biology, 2011). Sulonen and colleagues note in their conclusions: “An additional shortage is the lack of targeting of the 5' and 3' untranslated regions, especially in studies of complex diseases, in which protein coding sequences are not necessarily expected to be altered.”

In a more recent study, Wang and colleagues (PLoS ONE, 2018) when evaluating the WES gene coverage across TCGA exome kits mentioned that: *“However, most cases of undercovered genes were due to the inclusion of untranslated regions (UTRs). When only considering the coding sequences (CDS), only 2353 genes were observed to be undercovered”*

Regarding the provision of exact numbers for the STAD cohort, until quite recently, such an analysis could not be performed since Agilent was not providing the exact target baits and the TCGA kit was marked as “proprietary”. The recent release of the bait coordinates enabled us to calculate the exact coverage statistics:

The kit targets 33Mb of non-overlapping genomic regions, covering 44.91% of all nucleotides belonging to CDS-annotated regions, while the percentage drops to 0.31% when only 3'UTR regions are being evaluated.

This specific Agilent kit was used for 100% of the STAD TCGA cohort, and it's the most widely used exome kit in TCGA, applied on more than 6,000 samples. Similar results for a complete lack of UTR coverage have been reported for all the other WES TCGA kits since their bait coordinates were already available (Wang *et al.*, PLoS ONE, 2018)

A relevant mention has been added to the text:

“We now know that these regions are not covered in the TCGA WES kits ³⁰, with only 0.31% of 3'UTR regions being targeted in the TCGA STAD cohort”

And in the Methods section:

“The WES target baits utilized in the TCGA STAD study were downloaded from the location provided in the TCGA STAD patient manifest files.”

6) Figure 1B is highly dubious. Why is there a proximal accumulation of variants in the ICGC WGS, while there is an even distribution on the RNA-seq? This could be explained (perhaps) by the ICGC WGS data set being somatic (DNA) calls, while the RNA-seq calls are a combination of germline and somatic (DNA) + somatic (RNA editing). However, if this is the case, then this would logically indicate that the RNA-seq calls are largely dominated by germline calls given the uniformity. If the nature of the variant calls is suspect even in Figure 1, the entire paper is called into question.

We would like to clarify that in Figure 1b, as shown in the color legend, TCGA WES is the only dataset showing a proximal enrichment (color green), while ICGC WGS and RNA-Seq present a similar distribution across the 3'UTR length. This proximal enrichment, as also explained in the text, is due to the lack of targets in the TCGA WES in 3'UTR regions. The reads covering 3'UTRs are primarily from CDS fragment spillover. In order to further clarify this, we have changed the figure legend to mention:

“Fig 1b: Distribution of the relative position of single nucleotide variant calls along the 3' UTR as identified by analysis of TCGA WES, RNAseq and WGS data. TCGA WES calls are enriched proximally to the CDS region and depleted along the 3'UTR body due to the lack of 3'UTR targeting probes.”

7) It should be acknowledged in the Abstract that 60% of the variants are likely to be germline and of the remaining almost half (38-40%) are likely due to RNA editing, and that these are *lower limits*. The percentage of germline variants will likely increase if one further intersects those that have rs IDs. As a sanity check, the final set of variants that are “non-germline” and “non-RNA editing” should follow a curve similar to the ICGC WGS calls.

Following the Reviewer’s suggestion, we have edited the abstract to reference the percentages of germline variants and somatic mutations. Since the distinction between these categories took place using germline calls from blood (not predicted or based on rs IDs or gnomAD), we believe they are quite accurate. On the contrary, the potential RNA editing events we report are on the maximum limit since they are the product of overlap against external annotation and not the result of a sample-specific experiment.

The edited abstract now reads:

“Here, we identified common somatic and germline 3’ untranslated region (3’UTR) variants across the human transcriptome from 375 gastric patients from the The Cancer Genome Atlas (44.2% somatic).”

8) The three PD-L1 3’ UTR variants tested should be referred to a gene polymorphisms, not mutations. Please use this nomenclature throughout the manuscripts (polymorphisms refer to germline variants).

The text has been edited accordingly.

For instance, a relevant section now reads:

“We initially evaluated whether our high-throughput approach could capture the functional impact of the small number of 3’UTR SNVs that have been previously associated with changes in PD-L1 expression in gastric cancer, such as the polymorphisms: rs2297136²⁵ and rs4143815⁹², or in non-small cell lung cancer, such as rs4742098⁹³.”

9) For the subset of variants that are present in “5 or more samples, 058212 in the tested population”, what percentage falls into the germline category (see comment #5) and which somatic (DNA + RNA-editing)? I suspect that these will be enriched significantly above and beyond the 60% of germline variants. Here, my understanding is that the 60% figure highlighted in lines 320-322 is *before* the 5 or more samples threshold.

We would like to thank the Reviewer for this comment. As mentioned in Comment 1, we have added a detailed breakdown of the variants incorporated in the eQTL/ilQTL investigations as well. The relevant section now reads:

“We investigated variants that were present in 5 or more samples, corresponding to 1.3% or higher minor allele frequency (MAF) in the tested population (2,917,776 total variants, 68,4% / 1,994,516 germline, 31.6% / 923,260 somatic, in which 67.2% / 620,419 overlapping with editing sites).”

8) Somatic-only eQTLs are of interest (line 358-360) – can the authors highlight some examples?

More than 800K somatic eQTLs were identified, with many important eGenes identified. We have added some examples to showcase the breadth of the identified hits. The added text reads:

“Somatic eGenes, where the lead eQTL was somatic, included important gastric cancer oncogenes, such as KRAS, CCDN1, and CCND2 among others⁹⁹, genes involved in antigen generation, processing, or presentation (e.g. APOBEC3B, CANX, CTSS), and cytokines/chemokines or other key immune regulators (e.g. STAT1, CXCL5, CXCL9, TNFRSF9).”

9) How do the prioritized variants tested in the reporter assay map across the categories of germline/somatic(DNA)/somatic(RNA editing)?

Among the reporter assay variants, 478 were somatic (63.7%), while 135 (18%) overlapped with known locations for RNA editing events. A relevant mention has been added to the text

10) Is the association of variants with immune phenotypes (iQTLs) specific to immune phenotypes? How about other cancer hallmarks such as signatures of cell proliferation etc? The point that iQTLs appear to be somatic (lines 418-419) is of great interest – can the authors point to a few examples and show the evidence confirming the somatic nature?

We would like to thank the Reviewer for this comment. Indeed, the iQTLs are intrinsically specific to the immune phenotypes since the statistical model was specifically built to interrogate for variants exhibiting a significant dosage association with the TCGA iAtlas cross-modal immune phenotypes. The same approach and model could be deployed for the detection of other quantitative traits, including cell proliferation or specific hallmark signatures.

Regarding examples of somatic iQTLs, we added specific mentions as well as a note reminding the Readers that the vast majority of iQTLs are somatic.

The relevant section reads:

“Significant iQTLs, the majority of which of somatic origin, were identified in immune-relevant genes, including B2M, HLA genes, CANX, LDHA, PSMB2, and HNRNPR, which are known to affect the tumor immune landscape^{5,17,100-103}. The top hits also included WARS¹⁰⁴ and APOBEC3C¹⁰⁵, which were only recently implicated in tumor immunity, showing the potential of this approach for prioritization of novel cancer-specific immunotherapeutic targets.”

11) It should be acknowledged clearly (I would recommend at line 451) that the association of ADAR1 overexpression with ICI response is counter to the Manguso et al findings. This discrepancy should be discussed (in the Discussion). If the authors want to be ‘provocative’ (their own words), then they should be upfront about it.

We agree with the Reviewer’s suggestion and we have incorporated a more extensive discussion of the Manguso et al. findings as compared to our analysis.

We do not believe that our finding is directly counter to the Manguso, et al. findings. The Manguso study was performed solely in mice, and the main focus was to evaluate the effects of ADAR1 loss, especially in overcoming resistance to immunotherapy.

Our results align with the majority of published human study data, where baseline ADAR1 is identified as increased in responders to immunotherapy. The context and model used by Manguso are different from those of our investigation. To showcase this alignment, we meta-analyzed all melanoma studies available in the tumor immunotherapy gene expression resource (TIGER)[Chen et al., 2023, Genomics Proteomics Bioinformatics]. In the plot below, we see that ADAR1 baseline overexpression is a common feature for response to checkpoint inhibition in patients.

Fig. S6: ADAR overexpression is common in responders to immune-checkpoint inhibition.

Differential gene expression $\log_2(\text{fold change})$ (x axis) and $-\log_{10} \text{p-value}$ for baseline ADAR1 expression between responders and non-responders. $\log_2(\text{fold change}) = 0$, signifying equal expression between the two groups, has been marked with a vertical dashed line.

We have added the figure above as Supplementary Figure S6, as well as a more detailed commentary in the main text:

“By meta-analyzing additional studies through the tumor immunotherapy gene expression resource (TIGER), we see that ADAR1 overexpression is a common feature for response to checkpoint inhibition (Fig S6). On the other hand, Manguso and colleagues showed that loss of ADAR1 overcomes resistance to PD-1 checkpoint blockade caused by inactivation of antigen presentation by tumor cells in mouse models of resistance¹⁰¹.”

12) Some functional validation would be useful – for example, a knockdown of TARDBP and subsequent measurement of ADAR1 levels. Given the institutes the authors hail from, this should be absolutely trivial.

We would like to thank the Reviewer for this recommendation. We agree that further experiments could be useful in shedding light on the exciting TARDBP - ADAR1 connection identified through the variant analysis. However, we believe that they are beyond the scope of the present manuscript.

13) Can the authors test if the combination of iQTL PRS + PD-L1 improves ICI response prediction? Also is the power conferred by iQTL independent of PD-L1? This should be tested by multivariate analysis.

We would like to thank the Reviewer for this suggestion. The model combining the iQTL PRS and PD-L1 expression is indeed beneficial. In the derived multivariate linear model, both PD-L1 expression and iQTL presented as independent significant contributors (PD-L1 expression: coef. 0.02, $p=0.0136$, iLQTL PRS: coef. 0.05, $p=0.0038$), The combination presented the highest area under the receiver operating characteristic curve (AUC = 0.782) compared to each model separately.

The relevant text now reads:

“Importantly, the information captured by the PRS score is a predictor independent of PD-L1 expression, and their combination, as well as potentially the integration of the expression or mutational status of additional genes, can be leveraged to further increase the prognostic accuracy of the model (Fig. S9).”

Fig. S9: Comparison of iLQTL 3' UTR PRS with PD-L1 expression classification and a multivariate iLQTL + PD-L1 model in distinguishing responders and non-responders to ICI. Receiver Operator Characteristic (ROC) curves showing the ability of the PRS score, PD-L1 expression classification, and their linear combination with a regression model to distinguish between R and NR patients in the testing population (n=67). An Area Under the Curve (AUC) score is reported for all classifiers. The model combining the iLQTL PRS and PD-L1 expression presents the highest area under the receiver operating characteristic curve (AUC = 0.782), while both PD-L1 expression and iLQTL were identified as independent significant predictors (PD-L1 expression: coef. 0.02, p=0.0136, iLQTL PRS: coef. 0.05, p=0.0038).

Minor Comments

1) It should be acknowledged in the Introduction that RNA editing has a preference for UTR regions (PMID 22448268 and associated manuscripts)

We added a relevant mention about A-to-I editing in cancer and its significance and referenced:

1: ADAR1-Mediated RNA Editing and Its Role in Cancer, Liu *et al*, *Front Cell Dev Biol*, 2022

2: The effects of RNA editing in cancer tissue at different stages in carcinogenesis, Kurkowiak *et al*, *RNA Biol*, 2021

2) Some sentences are difficult to understand : “In PCAWG, RNAseq data analysis identified driver alterations in all 87 samples without a driver mutation detected at the DNA level” – all 87 samples out of how many samples? Without such details, it is not possible to evaluate such claims

We would like to thank the Reviewer for this comment. It is 87 samples out of all 87 samples without a driver mutation identified at the DNA level in the PCAWG study (100%).

The exact sentence from the referenced Nature manuscript is:

“Out of 87 samples from the PCAWG study that did not have a driver alteration at the DNA level and had RNA-sequencing (RNA-seq) data, every sample had an RNA-level alteration identified.”

We have edited our sentence to better match the original manuscript:

“In PCAWG (n=1,188), out of the 87 samples without a driver alteration identified at the DNA level and had available RNAseq data, every sample had an RNA-level alteration identified”

3) Shouldn't Figure 3C be normalized by genomic region length on the y-axis?

We would like to thank the Reviewer for the suggestion. We believe that a normalized graph is important in demonstrating the relative enrichment of 3'UTR significant cis-eQTLs and has therefore been added as Fig. S3C to complement Fig. S3B. Fig. S3C is important to demonstrate the number of 3'UTR significant cis-eQTLs, which is referenced several times in the manuscript.

Fig S3C. Rate per Mbp of significant cis-eQTLs in each genic region (5'UTR, CDS, 3'UTR). The rate was calculated by dividing the absolute number of significant cis-eQTLs mapping to each genic region divided by its collective genomic length.

4) How many iQTLs were identified? This number is not provided in the Results (line 413)

A total of 1,715 iQTLs were identified, with 159 (9.3%) identified as germline. Among the remaining 90.7% / 1,556 likely somatic iQTLs, 370 intersected with common RNA editing events.

The relevant mention has been added to the text.

Reviewer #5, expertise in cancer genomics, bioinformatics, eQTLs, non-coding mutations, TME and immune response to replace Reviewers #3 and #4 (Remarks to the Author):

After carefully reviewing the manuscript and the author's response to Reviewer #3 and Reviewer #4, my overall judgment is that the authors have mostly responded well to the questions. However, in some reasonable questions, particularly Reviewer #3: Major: 8, 9, 10, and Minor: 8, and Reviewer #4: Major 3, specific comments: Page 25, the authors did not make any substantial changes in the manuscript and their explanations were not very clear. Therefore, my suggestion is that the manuscript should be considered for publication only if the necessary revisions requested by Reviewer #3 and #4 are made. The following are my comments on the author's response to Reviewer #3 and #4.

We would like to thank the Reviewer for providing constructive feedback in order to improve this work as well as for recognizing that we have already responded well to most of the questions.

Reviewer #3:

Major:

Question 1: The authors responded well to this question. They clarified the discrepancy observed by the reviewer and explained that the 5,431,118 variants they identified are spread across the genome.

We thank the Reviewer for their comment. No additional edits or additions were made.

Question 2: The author provided a reasonable and acceptable response by pointing out that the lower number of eGenes found compared to some other studies is due to the heterogeneity in Calabrese et al.'s (2020) nature.

We thank the Reviewer for their comment. No additional edits or additions were made.

Question 3: The author correctly pointed out that the Griesemer study only utilizes germline variants and not somatic 3' UTR mutations. Therefore, as they claimed, it is indeed the first study to utilize a massively parallel reporter assay to evaluate somatic 3' UTR mutations.

We thank the Reviewer for their comment. No additional edits or additions were made.

Question 4: The author has now provided the permutation test results in the manuscript, which is necessary as suggested by the reviewer.

We thank the Reviewer for their comment. No additional edits or additions were made.

Question 5: Reviewer's question 5 is somewhat confusing, and the author has made their best effort to answer it. I believe their response is satisfactory.

We thank the Reviewer for their comment. No additional edits or additions were made.

Question 6: As the reviewer pointed out, the validation rate is not very high. However, considering the complexity of the experimental conditions and the cell lines used, it is understandable. Therefore, I think the authors have tried their best to answer this question and have done so properly.

We thank the Reviewer for their comment. No additional edits or additions were made.

Question 7: The authors have now rephrased the sentence in the manuscript for better understanding, which is acceptable.

We thank the Reviewer for their comment. No additional edits or additions were made.

Question 8: Reviewer #3 simply wants to know more about the TARDBP binding motif, and I think it is reasonable to ask this question. The author should at least discuss it in the discussion section.

We would like to thank the Reviewer for this suggestion. To enable the Readers further evaluate this we have:

i) Added all experimentally-derived RNA binding protein binding sites overlapping with the variant in the Supplementary Table S6. There, the binding site genomic location, the RNA binding protein, the experiment used for its identification, and related GEO datasets have been incorporated.

The relevant entry for the TARDBP binding site is:

chromosome	chr1
position	154583325
variant_id	chr1_154583325_T_C
chr_binding_site	chr1
start_binding_site	154583300
end_binding_site	154583330
Database	human_RBP_CLIPdb_2630536
strand	-
RBP	TARDBP
Type of evidence	PAR-CLIP,PARalyzer
Cell line	HEK293T
GSE study	DRA001158,DRS012387
Score	0.69584842

ii) We added in Table S6, a specific entry about the PAR-CLIP-defined TARDBP binding site overlapping the variant:

TARDBP Peak	Start: Chr1: 154583300	End: Chr1: 154583330	Peak Sequence / Binding Motif: GCTCTTGGAGTCATGACCAACACTCTAAAAG
----------------------------------	--------------------------------	--

iii) Updated the mention of the TARDBP binding, in order to provide further information about TARDBP, TARDBP binding preferences and motifs identified in the literature, the experiment used to identify this specific binding site, a reference to the information added in table S6, as well as relevant references from the literature.

The updated text reads:

“Based on a meta-analysis of RNA cross-linking and immunoprecipitation (CLIP) data from the POSTAR2 project⁵², the ADAR variant is predicted to overlap with multiple RBP binding sites (Table S6). One of those RBPs, TARDBP, has been studied for its ability to regulate gene expression, pre-mRNA editing, mRNA localization, and microRNA processing through binding on canonical GU-rich motifs or non-canonical sequences, with 3’UTRs being commonly targeted regions¹¹². TARDBP has been shown to directly regulate ADAR1 expression in liver cancer and leukemia cell line models¹¹³. Indeed, correlation analysis in gastric cancer patients from TCGA revealed a strong association in the expression of the two genes (Fig. 6C), suggesting that the regulation axis between TARDBP and ADAR could be functional in gastric cancer as well.”

Question 9: Reviewer #3 requested the authors to plot the results separately for both the melanoma and gastric adenocarcinoma cohorts. Although the authors plotted the results of gastric cancer in the response, they did not make any changes in the manuscript. I agree with Reviewer #3 that they should provide individual plots for these two cohorts in Figure 7a and Figure 6b. Alternatively, they should color the dots in Figure 6b and 7a to differentiate these two cancer types so that readers can understand that the difference between R and NR is independent of cancer type and not confounded by cancer types.

We would like to thank the Reviewer for this suggestion. Figures 6b and 7a have been updated accordingly.

Fig. 6B: A cohort of gastric cancer and melanoma patients was classified into R (responders) vs NR (non-responders) according to response efficacy with anti-PD-1 immunotherapy. Primary cancer RNAseq data from these patients were analyzed. Differential expression analysis revealed increased expression of ADAR

in R compared to NR. A Wilcoxon rank sum test was utilized to assess the significance of the comparison. Gastric and Melanoma cancer types are colored with grey and red, respectively.

Fig. 7A: iQTL polygenic risk score for predicting response to immunotherapy in melanoma and gastric cancer patients. (A) Comparison of the polygenic risk score (PRS) distribution in the non-responder (NR) and responder (R) groups of the testing population. A Wilcoxon rank sum test was utilized to assess the significance of the comparison. Boxplot lines represent the median and upper or lower quartiles, while upper whiskers represent the max and min. Gastric and Melanoma cancer types are colored with grey and red, respectively.

Question 10: The reviewer pointed out if the authors could compute the QTLs directly for R and NR as the phenotype in the training set and evaluate them in the test set. I think this is a good suggestion for the authors to try, and they should add the results at least in the supplemental material to compare with the results in Figure 7b. It is a good and not that complex approach for the authors to further support their claim that "this proof-of-concept is the first time that non-coding variants have been used to predict immunotherapy treatment outcomes in cancer."

We would like to thank the Reviewer for this comment. We would like to stress that the iQTLs used in the model have been calculated in TCGA and then directly applied to the training cohort based on the enrichment of the iQTLs in R/NR.

iQTL calculation requires large populations, and the training (n=68) and testing (n=67) populations are far underpowered to enable iQTL detection. Importantly, we followed the aforementioned approach in order to show that the iQTLs are not only functional but can be utilized across orthogonal studies without requiring each study to have the number of subjects or modalities able to support such investigations.

However, we identified variants enriched in 3'UTR or across the gene body in the training R and NR subgroups, where we calculated the PRS score directly for its ability to predict these outcomes without having to identify iQTLs. As the results show, iQTLs can generalize better in the testing population.

Fig. S10: Comparison of iQTL 3UTR PRS with PD-L1 expression and mutational burden in distinguishing responders and non-responders to ICI. Receiver Operator Characteristic (ROC) curves showing the ability of the PRS score classifications, calculated based on significantly top enriched variants from (i) iQTL (n=28, blue), (ii) 3' UTR (n=28, orange), (iii) all enriched variants (n=28, cyan), coupled with PD-L1 expression classification to distinguish between R and NR patients in the testing population (n=67). An Area Under the Curve (AUC) score is reported for all classifiers.

Minor:

Question 8: The reviewer pointed out that "ADAR appeared as a hotspot for 3' UTR eQTL...." This statement of "hotspot" should be supported with statistics, which I agree with. The authors should revise their claim in the manuscript accordingly.

The relevant sentence has been revised as follows:

"Moreover, ADAR harbored multiple 3'UTR eQTL and iQTL variants in gastric cancer"

Reviewer #4:

Major:

Question 1: The authors' response to this question was partially satisfactory. While I agree that trans effect analysis is beyond the scope of the present study, I believe they should at least discuss the possibility of a convergent function or mechanism of germline and somatic mutations in the 3' UTR of the ADAR gene, as suggested by the reviewer. This discussion would enhance the comprehensiveness of the manuscript.

We would like to thank the Reviewer for the comment. It is indeed an interesting question.

When examining 3'UTR eQTLs across all genes, we see that there is a very small overlap between germline and somatic hits. Specifically, 4.5% of somatic and germline eQTLs are located in the same microRNA binding site, and 5.3% for RBPs. In the case of ADAR, there are no somatic and germline eQTLs falling on the same regulatory microRNA or RBP binding site. This further strengthens the notion of functional divide that was discussed in the R1/Comment 2 and was also mentioned by Sheng and colleagues (Briefings in Bioinformatics, 2020).

The following text has been added to the manuscript:

“Around 90% of the 3'UTR eQTLs overlap with putative or experimentally-supported miRNA and RBP binding sites, providing a potential functional relevance for those variants, with only 1.6% and 1.83% of somatic and germline variants colocalizing on the same microRNA and RBP binding site, respectively.”

Question 2: The authors' response regarding the MPRA assay is reasonable. It is indeed a valuable approach to validate in silico predictions in a high-throughput manner. However, it is important to acknowledge that the validation rate is limited, which is also mentioned by Reviewer #3. The authors should address this limitation at least in the discussion.

We would like to thank the Reviewer for this suggestion. A relevant mention has been added to the discussion. The relevant passage reads:

“In addition to validating previously described 3'UTR eQTLs, our approach also identified novel 3'UTR variants and genes with immune-related functions in cancer. We utilized a massively parallel reporter assay (MPRA) to streamline validation across hundreds of candidate 3'UTR variants, with 15% exhibiting functional effects, even though the eQTLs were detected in patient samples and the MPRA assay was performed in gastric cancer cell lines, where the relevant RNA binding proteins, microRNAs and their targets might not exhibit conserved stoichiometry. The validation rate is comparable to the MPRA assay performed in Griesemer et al. ¹¹⁵”.

Question 3: I agree with Reviewer #4's suggestion to compare the performance of the proposed biomarkers with other well-known biomarkers such as tumor mutation burden and MSI. The references (77) for gastric cancer and (68) for melanoma contain the necessary data. For example, ref 77 dataset is available here: <https://www.ebi.ac.uk/ena/browser/view/PRJEB25780>. I am confused by the authors' claim that they are unable to obtain these metrics for comparison. They should be able to quickly compare the AUC from these data and include the results, at least in the supplemental material. Therefore, I strongly recommend that the authors perform the analysis as suggested by Reviewer #4.

We would like to thank the Reviewer for the comment. To clarify, not all WES datasets in the PRS study are readily available. We agree with the Reviewer that this specific subset (2 studies) can be utilized to perform the proposed analysis.

32 subjects out of 70 in these two studies were part of the PRS test set. We downloaded and analyzed the raw data using GATK best practices while the tumor mutational burden (TMB) and microsatellite instability (MSI) were calculated. iQTL Polygenic Risk Score presented the best performance with an AUC=0.904, compared to TMB and MSI with an AUC=0.642 and 0.517, respectively in these samples (**Fig. S8**).

Fig. S8: Comparison of iQTL 3' UTR PRS with PD-L1 expression classification, mutational burden and MSI in distinguishing responders and non-responders to ICI. Receiver Operator Characteristic (ROC) curves showing the ability of the TMB, MSI and PD-L1 expression classification to distinguish between R and NR patients in a subset of 32 patients from the PRS test set, where WES data were available. The Area Under the Curve (AUC) score is reported for all predictors and models.

The relevant text has also been added to the manuscript:

“When the selected variants were tested on the orthogonal test set (n=67), the polygenic risk score was significantly increased in the responders (Wilcoxon rank sum test p-value = 0.00071, Fig. 7A), and exhibited a higher area under the receiver operating characteristic curve (AUC, ROC) than PD-L1 expression (Fig. 7B), as well as against tumor mutational burden (TMB) or microsatellite instability (MSI) as calculated from WES data (Fig S8)”

Methods section

“Whole Exome Sequencing (WES) data for melanoma⁷⁰ and gastric cancer⁸⁰ were downloaded from the SRA using the sra-toolkit and processed according to GATK best practices using GATK 4.4⁸¹. Briefly, FASTQ files were checked for presence of contamination using FastQC 0.12.1⁸² and MultiQC 1.17⁸³, and following inspection, were aligned using the Burrows Wheeler Aligner (BWA) 0.7.17⁸⁴ using the BWA-MEM algorithm against the hg38 genome distributed by the GATK team

<https://console.cloud.google.com/storage/browser/genomics-public-data/resources/broad/hg38/v0>). The resulting SAM files were sorted & indexed using samtools 1.18⁸⁵. The files were then post-processed, marking duplicates and running Base Quality Score Recalibration (BQSR). A panel of normals was generated for each study using the healthy patient samples, which was then used along with each tumor-normal WES pair to call mutations using Mutect2, with gnomAD as a germline resource. To minimize artifact calls and contamination, the read orientation artifact workflow was followed before filtering the Mutect2 calls. To accelerate runtime, intervals were used according where available, utilizing the capture kit information for each study.

For the tumor mutational burden (TMB) calculation, SnpEff v5.2⁸⁶ with the GRCh38.105 database was used to annotate the resulting VCF files followed by TMB calculation using pyTMB v1.3⁸⁷ using a variant allele fraction (VAF) of 0.05, a mutation allele fraction (MAF) of 0.001, minimum depth of 20 and minimum alternative depth of 2 to minimize noise, while filtering out low quality, non-coding, synonymous and polymorphic calls against gnomAD⁸⁸. For the MSI calculation, MSIsensor-pro v1.2.0⁸⁹ was used by first scanning the reference genome for microsatellite information and then running in tumor-normal mode on the aligned reads with default parameters.”

Specific comments:

Page 25: The reviewer questioned how well these 28 genes are represented in the gastric cancer dataset. This is an important point to address, and the authors responded by stating that 15 out of the 28 variants were found in at least one sample. I believe this is crucial information that should be included in the main text of the manuscript.

We would like to thank the Reviewer for their recommendation. We have added a relevant mention in the Results section.

The text now reads:

“The score is calculated as the number of variants detected in the patient’s tumor sample, therefore ranging from 0 to 28. All variants were present across the cohort in 1 or more individuals, with 15 being present in the smaller gastric cancer sub-population (n=45).”

Other minor questions that I believe should be addressed:

1. In Figure 5, the figure legend should mention a dark line instead of a dashed red line.

The mention has been corrected

2. In Figure 3a, the author only provides a very rough formula for eQTL and doesn't even include the genotype in the formula. I suggest they provide a detailed formula in the methods section.

The detailed formula has been included in the methods section.

We sincerely believe that the comments strengthened the manuscript and enabled additions that would be useful to the Readers. We would like to thank the Reviewer for their comments and suggestions.

Reviewers' Comments:

Reviewer #1:

Remarks to the Author:

The authors have done a good job in addressing the previous concerns.

Reviewer #5:

Remarks to the Author:

The authors have addressed all the comments, no further questions.

Response to Reviewer Comments

“Reviewer #1 (Remarks to the Author):

The authors have done a good job in addressing the previous concerns.

Reviewer #5 (Remarks to the Author):

The authors have addressed all the comments, no further questions.”

We thank the reviewers for the previous comments.